**Reversible and irreversible gas-particle partitioning of dicarbonyl compounds observed in the real atmosphere**

Jingcheng Hu[1], Zhongming Chen[1], Xuan Qin[1], and Ping Dong[1]

[1]State Key Laboratory of Environmental Simulation and Pollution Control, College of Environmental Sciences and Engineering, Peking University, Beijing, 100871, China

*Correspondence to*: Zhongming Chen (zmchen@pku.edu.cn)

**Abstract.** Glyoxal and methylglyoxal are vital carbonyl compounds in the atmosphere and play substantial roles in radical cycling and ozone formation. The partitioning process of glyoxal and methylglyoxal between the gas and particle phase via reversible and irreversible pathways could efficiently contribute to secondary organic aerosol (SOA) formation. However, the relative importance of two partitioning pathways still remains elusive, especially in the real atmosphere. In this study, we launched five field observations in different seasons and simultaneously measured glyoxal and methylglyoxal in the gas and particle phase. The field-measured gas-particle partitioning coefficients were 5–7 magnitudes higher than the theoretical ones, indicating the significant roles of reversible and irreversible pathways in the partitioning process. The particulate concentration of dicarbonyls and product distribution via the two pathways were further investigated using a box model coupled with the corresponding kinetic mechanisms. We recommended the irreversible reactive uptake coefficient γ for glyoxal and methylglyoxal in different seasons in the real atmosphere, and the average value of $8.0 \times 10^{-3}$ for glyoxal and $2.0 \times 10^{-3}$ for methylglyoxal best represented the loss of gaseous dicarbonyls by irreversible gas-particle partitioning processes. Compared to the reversible pathways, the irreversible pathways played a dominant role, with a proportion of more than 90% in the gas-particle partitioning process in the real atmosphere and the proportion was significantly influenced by relative humidity and inorganic components in aerosols. However, the reversible pathways were also substantial, especially in winter, with a proportion of more than 10%. The partitioning processes of dicarbonyls in reversible and irreversible pathways jointly contributed to more than 25% of SOA formation in the real atmosphere. To our knowledge, this study is the first to systemically examine both reversible and irreversible pathways in the ambient atmosphere, strives to narrow the gap between model simulations and field-measured gas-particle partitioning coefficients, and reveals the importance of gas-particle processes for dicarbonyls in SOA formation.

**1 Introduction**

Glyoxal and methylglyoxal, the simplest α-dicarbonyls, are recognized as being of great importance in atmospheric chemistry due to their unique physicochemical properties. The α-dicarbonyl functionality leads to higher water solubility and reactivity

of dicarbonyls than expected, as the α-dicarbonyl functionality is hydrophilic and contributes to hydrate formation. The hydrate form of carbonyls is less volatile and more water-soluble than the unhydrated form (EPA., 2012), owing to the strong effect of the two hydrogen-bonding groups in the hydrated form (Elrod et al., 2021). Moreover, hydrates can easily participate in continuous radical reactions with higher reactivity by H-abstraction to form higher-molecular-weight oligomers (Michailoudi et al., 2021).The traditional opinion is that methylglyoxal is less reactive compared to glyoxal due to its unreactive methyl substitution, while a very recent study noted that methylglyoxal could be more reactive under an atmospheric-relevant concentration (Li et al., 2021). Overall, both of them play crucial roles in radiation balance, air quality, brown carbon formation, and SOA formation (Laskin et al., 2015; Qiu et al., 2020). Moreover, as major carcinogenic and genotoxic compounds, dicarbonyls can cause serious damage to human health. They have relatively limited primary sources, except for biomass burning and biofuel combustion (Zarzana et al., 2017; Zarzana et al., 2018), compared to secondary formation that occurs with photooxidation of both biogenic volatile organic compounds (VOCs), such as isoprene, and anthropogenic VOCs, such as aromatic hydrocarbons (Lv et al., 2019). Considering the atmospheric sink, glyoxal and methylglyoxal can be lost in the gas phase by self-photolysis, oxidation by active radicals (such as OH radicals, $NO_3$ radicals) and wet/dry deposition; however, there is still a missing sink for the two dicarbonyls (Volkamer et al., 2007), that's the gas-particle partitioning process, which would be fully discussed in this study.

Gas-particle partitioning was recently found to be the most important removal pathway for both glyoxal and methylglyoxal, especially in regions like Beijing with high particulate matter (PM) pollution that provides sufficient aerosol surface area. Although they have relatively high vapor pressure, glyoxal and methylglyoxal can efficiently partition into the particle phase due to their α-dicarbonyl functionality. The surface-adsorbed dicarbonyls could alter the properties of the particle's surfaces and the organic surface films could act as a kinetic barrier to gas-aerosol mass transport and thereby influence particle equilibration and water/gas uptake (Donaldson and Vaida, 2006). Upon physical adsorption, besides desorption or reaction at the surface, dicarbonyls could undergo solvation and incorporation into the bulk liquid, and then they could go through diffusion and chemical reactions in the bulk phase. The product may return into surfaces and gas phase, or stay in the bulk phase (Paul et al., 2011). Moreover, chemical reactions occurred at the surface or in the bulk phase could in turn accelerate the physical adsorption and greatly contribute to the formation and growth of atmospheric particulate matter. Whereas, as it is difficult to distinguish the surface reactions and bulk reactions in field observations, we regard both of them as particle-phase reactions in this study. The chemical reactions occurring in the gas-particle partitioning processes can be divided into reversible pathways, including reversible hydration and self-oligomerization, and irreversible pathways, which can be driven by oxidative compounds. These processes can also efficiently explain observed aerosols properties – including relatively high oxygenation levels, compositions such as organic acids and oligomers, and higher light absorption – that cannot be explained by traditional absorptive models of gas-particle partitioning (Pankow, 1994; Pankow and James, 1994; Odum et al., 1996).

Many laboratory and model studies have made a great effort to investigate the reversible and irreversible pathways of dicarbonyls to further understand their gas-particle partitioning mechanisms and reveal their contribution to SOA formation. Fu et al. (2008) found that the modeled SOA concentrations were largely increased when accounting for irreversible uptake of dicarbonyls in the GEOS-Chem model. Considering the surface-controlled reactive uptake of dicarbonyls into the CMAQ model, the aerosol uptake of dicarbonyls accounted for more than 45% of total SOA in the eastern US (Ying et al., 2015); similarly, the contribution of glyoxal and methylglyoxal to SOA formation in China was 14% to 25% and 23% to 28%, respectively (Hu et al., 2017). Although reversible and irreversible pathways of dicarbonyls have been separately investigated in previous studies, solely incorporating just one pathway into models could lead to a large discrepancy between model results and observational data, highlighting the importance of comprehensively considering both reversible and irreversible pathways when quantifying the gas-particle partitioning process of dicarbonyls (Li et al., 2014; Hu et al., 2017; Ling et al., 2020).

Despite increasing interest in dicarbonyls and their gas-particle partitioning processes, the detailed chemical mechanisms of two partitioning pathways remain poorly understood. First, previous studies have exposed seed particles to high concentration levels of dicarbonyl vapors, from hundred ppb to ppm levels, or used bulk samples; thus, their applicability to the real atmosphere requires further validation. Second, prior studies always used one constant coefficient to present all heterogeneous processes occurring on the aerosol, which neglects the influencing factors in real atmospheric partitioning processes. Further studies have shown that the two pathways in the gas-particle partitioning process for glyoxal and methylglyoxal are rather complex, and their relative contribution to the partitioning process can be influenced by many factors such as relative humidity (Curry et al., 2018; Shen et al., 2018), particle acidity (Liggio et al., 2005b; Shi et al., 2020), and particle organic/inorganic components (Kampf et al., 2013). However, there persist controversies in the specific partitioning mechanisms of glyoxal and methylglyoxal, especially conflicting views on their role in SOA formation, which urgently warrants further investigation.

In this study, five field observations were launched over urban Beijing in four seasons, and glyoxal and methylglyoxal in the gas and particle phase were simultaneously measured. Beijing, as the political center of China, is the most prosperous city with numerous key environmental issues. Chen et al. (2021) found that the average concentration of dicarbonyls in Beijing is lowest among the key regions that have relatively higher $PM_{2.5}$ concentrations, indicating there is a more efficient partitioning process of dicarbonyls. Thus, it is more environmentally significant to discuss the gas-particle partitioning processes in urban Beijing. These processes are divided into reversible pathways and irreversible pathways, which is based on the reversibility of chemical reaction of dicarbonyls occurred on condensed phase (Galloway et al., 2008; Ervens and Volkamer, 2010; Kampf et al., 2013; Ling et al., 2020). On the basis of field-measured data, we could estimate the product distribution, main influencing factors, and relative importance of the two gas-particle partitioning pathways for glyoxal and methylglyoxal in the real atmosphere.

**2 MATERIALS AND METHOD**

## 2.1 Field sampling

We performed field observations on the roof of a six-story teaching building (26 m above the ground) on the Peking University campus (39.992°N, 116.304°E) in northwest urban Beijing. The field observations in this study were launched during four different seasons from 2019 to 2021.

Gaseous carbonyls were collected by adsorption reactions in a 2,4-dinitrophenyl hydrazine (DNPH) cartridge (Sep-Pak; Waters Corporation). The air samples were first passed through an ozone scrubber (Sep-Pak; Waters Corporation) to eliminate interference by ozone and then trapped in the DNPH cartridge. To prevent deliquescence of the potassium iodide in the ozone scrubber, the air samples were mixed with ultrapure nitrogen before pumped into the sampling tubing. Air samples were continuously collected every 3 h in daytime and 9 h in nighttime. The total flow rate was 0.8 L·min$^{-1}$.

Particulate carbonyls were collected by a four-channel ambient particles sampler (TH-16A, Wuhan Tianhong) with Teflon filter and quartz filter (47 mm, Whatman). The Teflon filter was used to measure the mass concentration of collected PM$_{2.5}$ and water-soluble inorganic compounds (Na$^+$, NH$_4^+$, K$^+$, Mg$^{2+}$, Ca$^{2+}$, Cl$^-$, NO$_3^-$, and SO$_4^{2-}$). The quartz filters were used for carbonyl analysis. The flow rate was set at 16.7 L·min$^{-1}$ and particle samples were continuously collected every 12 h daily. Detailed information about field sampling and analysis were provided in the previous studies (Rao et al., 2016; Qian et al., 2019). To estimate the positive artifacts by adsorption of gas-phase dicarbonyls onto the filter (Hart and Pankow, 1994; Mader and Pankow, 2001; Liggio, 2004; Odabasi and Seyfioglu, 2005), throughout our previous field observations, we placed a backup quartz filter after the particle sampling quartz filter using an independent filter holder. The sampling filters would collect the particles and adsorbed gaseous dicarbonyls, while the backup filter would only collect gaseous dicarbonyls. And the ratio of measured dicarbonyls in second filter to that in the first were lower than 20%, which was equal to the previous study (Shen et al., 2018). And the particulate concentrations of dicarbonyls used in this study were already corrected by the possible adsorption artifacts.

The meteorological station was co-located at our sampling site and provided meteorological parameters. Common trace gases, such as NO/NO$_2$, SO$_2$, CO, and O$_3$, were detected online by Thermo 42i, 43i, 48i, and 49i analyzers, respectively. A TEOM 1400A analyzer was applied to measure the mass concentrations of PM$_{2.5}$ and PM$_{10}$, the results of which were consistent with the PM$_{2.5}$ weighing results (Fig. S1). The time resolution for all of the above data was 1 min. Detailed information about these five observations is shown in Table S1.

## 2.2 Sample extraction and analysis

The gaseous carbonyl samples were eluted with acetonitrile (HPLC/GC-MS grade) at a flow rate of less than 3 mL/min (higher flow rates can result in reduced recovery). And the particulate carbonyl samples on quartz filter were eluted with acidic DNPH solutions in the flask and then were shaken for 3 h at 4 °C with a rotation rate of 180 rpm in an oscillator (Shanghai Zhicheng

ZWY 103D). The derived solutions were placed in darkness for 12-24 h to ensure complete derivatization, and then they were analyzed by high-performance liquid chromatography-ultraviolet (HPLC-UV) for separation and detection. Carbonyls were separated effectively with each other (Fig. S2) in this method. They were calibrated using a mixing standard solution with a concentration range of 0.1–10 μM, and the linearity was indicated by a correlation of determination ($r^2$) of at least 0.999. The detailed analysis method was presented in the previous study (Wang et al., 2009).

The Teflon samples were also extracted by deionized water using an ultrasonic bath for 30 min at room temperature. The extracted solutions were analyzed by ion chromatography (IC Integrion and Dionex ICS 2000, USA) to measure the water-soluble inorganic compounds ($Na^+$, $NH_4^+$, $K^+$, $Mg^{2+}$, $Ca^{2+}$, $Cl^-$, $NO_3^-$, and $SO_4^{2-}$) and low-molecular-weight organic acids (formate, acetate, and oxalate) in aerosols.

**2.3 Quality assurance / quality control**

As carbonyl compounds are ubiquitous in envireonmental media, following measurements were conducted during the sample collection, pretreatment and analysis to ensure the accuracy of results: (1) Before sampling, flow calibration and airtightness tests of sampling devices were conducted, and flow difference were less than 10%; (2) After sampling, the gas-phase samples were resealed by its end cap and plug, and stored in the provided pouch under cool environment (<4℃); the particle-phase samples were stored in the sealed boxes wrapped by pre-baked aluminum foils under freezing environment (<-18℃), both gas-phase and particle-phase samples were extracted and analyzed within a week; (3) The extraction processes were conducted in fume hood with glassware, which was rinsed with acetonitrile for at least three times; (4) A calibration run was performed each day to determine the response factor of the detector and recalibration was performed if the relative deviation of the RF was beyond 5%.

Blank samples were collected every three days and then were stored and extracted by the same procedure as that for ambient samples. The blank gas-phase samples were collected by placing blank DNPH cartridges near the gas inlet for the same duration without artificial pumping. And the blank particle-phase samples were collected by placing blank quartz filter on the $PM_{2.5}$ inlet with flow rate of 0 L/min. All data used in this study were all calibrated by blanks.

The limit of detection (LOD) of two methods was 50 pptv for gaseous carbonyls and 1 ng·m⁻³ for particulate carbonyls, which is similar to the previous literature (Shen et al., 2018). Sample amount to limit of detection ratios were significantly higher than 1.0 for both gas- and particle-phase samples, indicating that the sensitivity of the methods was sufficient to analyze the samples.

Additional field-sampling were launched to estimate the sampling efficiency during the collection. Two blank DNPH cartridges were connected in tandem to assess the sampling efficiency of gas-phase carbonyls. The sampling efficiency of the cartridges were the ratios of dicarbonyl concentrations in the first cartridge to the total concentrations in the two cartridges and the results

were more than 95% for both glyoxal and methylglyoxal. Similarly, a backup Teflon filter were placed after the particle

sampling Teflon filter using an independent filter holder to estimate the particle collection efficiency. Both Teflon filters were

weighed by a semimicro balance (Sartorius, Germany) to obtain the mass concentration of collected particles. The mass

concentrations of particles collected on the backup filter were closed to zero, indicating that the sampling efficiency of particle

were more than 99%.

Moreover, recovery tests were also conducted using two methods - adding standard solution and repeated extraction. We added

the standard solution at three spiked levels of 0.025, 0.25 and 2.5 µg (namely 50 µL of 0.5 µg·mL-1, 5 µg·mL-1 and 50 µg·mL-

1 analytical standards) into blank DNPH cartridges and blank quartz filter to determine the carbonyl lost during the extraction

and derivation. And then the cartridges and filters were extracted in the same way as the ambient samples. Each group were

set with five parallel. The recoveries were ranged from 88% to 96% for gas-phase method and ranged from 85% to 96% for

particle-phase method. Moreover, we also estimated the recovery efficiencies by repeated extraction and the recoveries were

the ratios of dicarbonyl concentrations in the first extraction to the total concentrations in the two extractions. The results

ranged from 92.8% to 99.9% for gas-phase method and ranged from 90% to 99.9% for particle-phase method.

**2.3 Estimation of effective partitioning coefficient**

To estimate the effective partitioning process of gas-phase carbonyls to the particle phase, we could use Pankow's absorptive

partitioning theory for the gas-organic phase (Eqs. (1), (2)) (Odum et al., 1996) and Henry's law for the gas-liquid phase (Eq.

(3)):

$$K_p^f = \frac{C_p}{C_g \times TSP} \tag{1}$$

$$K_p^t = \frac{RTf_{om}}{10^6 MW_{OM} \zeta p_L^0} \tag{2}$$

$$eff\ K_H = 10^3 \frac{c_p}{c_g \times M \times ALWC/\rho_{water}} \tag{3}$$

In Eq. (1), $K_p^f$ (m$^3$·µg$^{-1}$) is the field-measured gas-particle partitioning coefficient; $C_p$ (µg·m$^{-3}$) is the concentrations of

dicarbonyls in the particle phase, which is derived from the analysis of extracts, including monomers and their reversibly

formed products (the product distribution is discussed in Section 3.2); $C_g$ (µg·m$^{-3}$) is the concentration of dicarbonyls in the

gas phase; and TSP (µg·m$^{-3}$) is the mass concentration of suspended particles (mass concentrations of PM$_{2.5}$ were used in this

study). In Eq. (2), $K_p^t$ (m$^3$·µg$^{-1}$) are the theoretical gas-particle partitioning coefficients determined by Pankow's absorptive

model, $f_{om}$ is the absorbing fraction of total particulate mass, MW$_{OM}$ (g·mol$^{-1}$) is the mean molecular weight of the organic

phase, and $\zeta$ is the activity coefficient of target compounds. In the estimation of $K_p^t$ in this study, $f_{om}$ and $\zeta$ are unity and MW$_{OM}$

= 200 g·mol$^{-1}$, as used in previous studies (Healy et al., 2008; Williams et al., 2010; Xie et al., 2014; Shen et al., 2018), and

$p_L^0$(Pa) is the supercooled vapor pressure of compounds as a pure liquid at temperature T, which is calculated by the extended

aerosol inorganic model (E-AIM, http://www.aim.env.uea.ac.uk/aim/ddbst/pcalc_main.php) (Clegg et al., 1998). The possible

uncertainty in $K_p^t$ calculation were fully discussed in Supporting Information. In Eq. (3), eff $K_H$ (M·atm$^{-1}$) is the field-derived

effective Henry's law coefficient; $c_p$ (μg·m$^{-3}$) and $c_g$ (atm) are particle- and gas-phase concentrations of carbonyls, respectively;

ALWC (μg·m$^{-3}$) is the aerosol liquid water content calculated by the thermodynamic model ISORROPIA-II(forward model,

metastable state), the results of which are comparable to the actual measured contents confirmed by previous studies (Guo et

al., 2015).

The irreversible reactive uptake coefficient γ could efficiently describe the irreversible pathways of the gas-particle partitioning

process of dicarbonyls driven by OH radicals. We could estimate the reactive uptake coefficients γ based on the effective

Henry's constant via theory calculation (Hanson et al., 1994; Curry et al., 2018) and then calculate the effective uptake rate

$k_{eff, uptake}$, following Eqs. (4)–(7):

$$\frac{1}{\gamma} = \frac{1}{\alpha} + \frac{v}{4RT\,eff\,K_H\sqrt{k^l D_{aq}}} \times \frac{1}{[cothq-1/q]} \tag{4}$$

$$v = \sqrt{\frac{8RT}{\pi M_X}} \tag{5}$$

$$q = R_p/l = R_p/\sqrt{\frac{D_{aq}}{k^l}} \tag{6}$$

$$k_{eff,\,uptake} = \frac{1}{4}v \times \gamma \times A_{surf} \tag{7}$$

where γ is the dimensionless uptake coefficient, v (m·s$^{-1}$) is the gas-phase thermal velocity of glyoxal/methylglyoxal, $D_{aq}$

(m$^2$·s$^{-1}$) is the diffusion coefficient in the liquid phase, α is dimensionless mass accommodation coefficient, eff $K_H$ (M·atm$^{-1}$)

is the effective Henry's law constant calculated by field-measured data in Table 2, R is the universal gas constant, $k^l$ (s$^{-1}$) is the

first-order aqueous loss rate, $M_X$ (kg·mol$^{-1}$) is the average molar mass of gas-phase dicarbonyls, q is the parameter for

measuring in-particle diffusion limitations, $R_p$ (m) is the particle radius, l (m) is the diffusion reactive length, $k_{eff, uptake}$ (s$^{-1}$) is

the effective uptake rate, and $A_{surf}$ (m$^2$·m$^{-3}$) is the aerosol surface area density. This formulation is based on the effective Henry's

law constant under high RH conditions (RH > 40%). Moreover, the formulation describes the reactive uptake due to irreversible

multiple-phase loss processes in the presence of OH. The uncertainty in the γ calculation is mainly attributed to the uncertainty

in OH concentration, which was $3 \times 10^{-12}$ M on average and varied from $5.5 \times 10^{-14}$ to $8 \times 10^{-12}$ M (Herrmann et al., 2010).

**3 RESULT AND DISCUSSION**

**3.1 Observation results and partitioning coefficients calculation**

**3.1.1 Dicarbonyls in the gas and particle phase**

We launched five field observations in different seasons. Table S1 details the information about the field observations,

including observation periods, sample volume, and meteorological parameters. We totally collected 387 gas-phase samples

and 130 particle-phase samples in four seasons. In these samples, carbonyls were simultaneously measured in both gas phase and particle phase. Ten carbonyls were measured in the gas phase, including formaldehyde, acetaldehyde, acetone, propionaldehyde, methacrolein, butyraldehyde, methyl vinyl ketone, benzaldehyde, glyoxal, and methylglyoxal; and six carbonyls were measured in the particle phase, including formaldehyde, acetaldehyde, acetone, propionaldehyde, glyoxal, and methylglyoxal. In this study, we mainly discuss the gas-particle partitioning processes of glyoxal and methylglyoxal because of their significant roles in atmospheric chemistry.

Figure 1 and Table 1 show the temporal characteristics and seasonal variation of glyoxal and methylglyoxal, respectively. Gaseous dicarbonyls showed obvious seasonal variation. Concentrations in summer ($0.99 \pm 0.59$ ppbv) were generally much higher than in other seasons, followed by autumn and spring, and the concentrations in winter were the lowest. This seasonal variation could be partly attributed to the higher temperature and more intensive radiation in summer, which could greatly enhance the secondary formation of gaseous carbonyls via photochemical reactions. The diurnal variation in the dicarbonyls during summer support this interpretation of the data; gas-phase dicarbonyls exhibited obviously diurnal variations in summer, whereas this variation was irregular in other seasons (Fig. S3). The concentration levels of gaseous dicarbonyl in summer rapidly increased after sunrise, remained relatively high during the daytime (12:00–14:00), and then decreased at dusk. Although methylglyoxal has a shorter lifetime compare to glyoxal (glyoxal 2.9 h vs. methylglyoxal 1.6 h) (Fu et al., 2008), its gas-phase concentration levels were generally higher than those of glyoxal, consistent with previous studies (Rao et al., 2016; Mitsuishi et al., 2018; Qian et al., 2019), mainly due to the relatively larger production from isoprene and acetone for methylglyoxal.

The concentrations of particulate dicarbonyls were an order of magnitude smaller than the gaseous concentrations using the unit of nanogram per cubic meter of air (ng/m$^3$ air). The average particulate glyoxal and methylglyoxal were 19.37 and 11.24 ng/m$^3$, respectively, which were slightly higher than previously reported values (Zhu et al., 2018; Shen et al., 2018; Qian et al., 2019; Cui et al., 2021). Dicarbonyls measured in the particle phase also showed obvious seasonal variation. The particulate concentrations of the two dicarbonyls in winter ($43.38 \pm 32.42$ ng/m$^3$ air) were 2–2.3 times higher than those in other seasons, suggesting that the dicarbonyls were more favored into the particle phase in winter. Moreover, particulate dicarbonyls in different seasons exhibited the same diurnal variation (Fig. S3). The particulate concentrations of dicarbonyls in daytime were generally higher than those in nighttime, especially in winter.

### 3.1.2 Gas-particle partitioning coefficient

Dicarbonyls could partition between gas and aerosol phases or the liquid phase, following Pankow's absorptive partitioning theory or Henry's law, respectively, as listed in Table 2. Both gas-particle partitioning coefficient ($K_p^f$) and effective Henry's law coefficient ($K_H^f$) were calculated on the basis of field-measured data and were in the range of $10^{-4}$–$10^{-2}$ m$^3 \cdot \mu$g$^{-1}$ and $10^6$–

$10^8$ M·atm$^{-1}$, respectively. The partitioning coefficient values of the two dicarbonyls exhibited the same seasonal variation, as

winter and spring > autumn > summer. A higher aerosol concentration accompanied by higher aerosol surface area

concentration and lower relative humidity resulted in a higher partitioning coefficient in winter and spring, when heavy

pollution and sandstorms always occurred. In the case of temperature variation varied from 265.53 K to 310.75 K in different

seasons, lower temperature promoted the gas-to-particle partitioning processes as $K_p^f$ values for the dicarbonyls and

temperature showed negative correlation with significant difference ($p < 0.001$) (Fig. S4). Moreover, The $K_p^f$ and $K_H^f$ values

of glyoxal were always higher than those of methylglyoxal, implying the former was more likely to partition to the particle

phase; this could be attributed to their different structures. Glyoxal were more soluble and reactive because of the adjacent

electron-poor aldehydic carbons, whereas methylglyoxal was more stable due to the reduced electron-deficient ketone moiety

(Kroll et al., 2005).

Both $K_p^f$ and $K_H^f$ were relatively close to those found in previous field-measured studies (Shen et al., 2018; Qian et al., 2019;

Cui et al., 2021). However, compared with the theoretical partitioning coefficients $K_p^t$ calculated by Pankow's absorptive

theory, $K_p^f$ values were approximately 5–7 orders of magnitudes higher than the corresponding $K_p^t$ values. Similarly, $K_H^f$ values

were approximately 2–5 orders of magnitudes higher than the theoretical Henry's law coefficient $K_H^t$ calculated in pure water,

which could be attributed to salting effects in wet aerosol (Figure S5). $K_p^f/K_p^t$ values in this study were close to but slightly

higher than the values published in previous literature (Table S2) and the discrepancy between field-measured partitioning

coefficients and the theoretical ones was fully discussed in Suppoting Information.

To narrow the large discrepancy between field-measured partitioning coefficients and theoretical ones, we needed to further

investigate the mechanism and product distribution of chemical reactions occurring in the aerosols during the partitioning

processes. The products of the reversible and irreversible pathways mostly have lower saturated vapor pressure, and thus

leading to higher partitioning coefficients compared to monomer dicarbonyls. Take glyoxal for example, the effective

saturation vapor pressures of the product set in reversible pathways are ~$10^{-5}$ Torr in the real atmosphere (Shen et al., 2018).

And the products of the irreversible pathways had much lower vapor pressure values than those of reversible pathways, for

example, the vapor pressure of oxalic acids and ammonium oxalates are ~$10^{-5}$ Torr (Saxena and Hildemann, 1996) and $5.18 \times$

$10^{-8}$ Torr (EPA, 2011), respectively, and those of glyoxal trimer dihydrates are ~$10^{-11}$ Torr at 20 °C (SPARC, 2003), indicating

the irreversible pathways make larger contributions to the underestimation of partitioning processes of dicarbonyls. The

following sections further discuss the mechanism and product distribution of reversible and irreversible pathways to explain

the partitioning process of dicarbonyls.

**3.2 Reversible pathways**

Gas-particle partitioning of dicarbonyls via reversible pathways mainly consists of hydration and self-oligomerization. Since

glyoxal and methylglyoxal had high water solubility and reactivity, they could easily dissolve into aerosol liquid water, and then form hydrates and oligomers. Hemiacetal/acetal formation (Loeffler et al., 2006) and aldol condensation (Haan et al., 2009) are the most thermodynamically favored oligomer reactions for glyoxal hydrates and methylglyoxal hydrates, respectively. The proposed mechanism for the reversible formation of glyoxal and methylglyoxal in aerosols is shown in Fig. S6. By adding excess derivatization agent (like 2,4-dinitrophenylhydrazone in this study), dicarbonyls as well as their reversibly formed products are efficiently transformed into dicarbonyl-bis-2,4-dinitrophenylhydrazone, which are quantified as monomers by means of analysis techniques (Kampf et al., 2013). Moreover, Healy et al. (2008) have confirmed that derivatization agent was found to efficiently dissolve a trimeric glyoxal standard and convert the resulting monomers to oxime derivatives, and oligomers could not be detected in the extracts of filter samples by GC-MS analysis, also indicating the use of excess derivatization agent could efficiently convert the hydrates and oligomers back to the monomeric species by removing dicarbonyl monomers from the extract as soon as they are formed. Both dissolves dicarbonyl monomers and reversibly formed production are efficiently transformed into carbonyl-bis-2,4-dinitrophenylhydrazone, which was quantified by means of HPLC-UV in this study. The concentrations of dissolved dicarbonyl monomers were estimated using Henry's law coefficients, which is used to determine the physical solubility of carbonyls (e.g., $K_H$=5 M·atm$^{-1}$ for glyoxal) (Schweitzer et al., 1998). The results were negligible compared to the concentrations of carbonyls in hydrate and oligomer forms. Thus, the concentrations of particle-phase dicarbonyl in reversible partitioning pathways were close to the measured concentration of carbonyls by HPLC-UV.

As glyoxal and methylglyoxal have similar trend under different conditions, we focused on the total concentration of the two dicarbonyls in the following discussion. As shown in Fig. 2a, the particulate concentration of dicarbonyls via a reversible pathway was strongly dependent on RH. It increased significantly when RH increased from <10% to 60%, as dicarbonyls were more favorable to dissolve into hygroscopic aerosols during their growth (Mitsuishi et al., 2018; Xu et al., 2020). However, from 60% to 80% RH, it exhibited the opposite trend and decreased with increasing RH, as higher water concentrations at elevated RH levels may dilute the monomer concentration in the particle phase and hinder oligomerization reactions (Healy et al., 2009), and the product distribution of the reversible formation could also well explain this phenomenon. The results exhibited similar partten to a previous study, in which the partitioning of glyoxal and methylglyoxal gradually increased as RH increased to 40%, peaked sharply around 50%, and subsequently decreased as RH increased towards 80%(Healy et al., 2009). Moreover, ionic strength could also influence the reversible partitioning process as it is closely related to aerosol liquid water and RH conditions. The presence of inorganic ions could catalyze and participate in oligomerization reactions via salting effects (Sareen et al., 2010; Mcneill, 2015). Whereas, increasing viscosity of particles with increasing ionic strength could slow down all particle-phase reactions, and the reversible nucleophilic addition of inorganic ions (e.g., sulfate ions) at carbonyl carbons deactivates the molecule for further oligomerization (Kampf et al., 2013).

To roughly estimate the product distribution of the reversible pathway in the real atmosphere, we simplified reaction

mechanisms and calculated the product distribution on the basis of the kinetic mechanisms listed in Table S3 using a 0-D box

model with a steady-state approach. Generally, more dicarbonyls existed in oligomer forms than in hydrate forms in the

reversible formation. Moreover, their distribution exhibited obvious seasonal variations. Summer had the highest proportion

of hydrate forms, while winter had the highest proportion of oligomer forms. Detailed information is shown in Table S4. The

seasonal variation could be attributed to the RH in different seasons – relatively high in summer and low in winter. As shown

in Fig. 2b, the product distribution of the reversible formation has a strong dependence on RH. The proportion of dicarbonyls

in hydrate forms increased with increasing RH and could reach more than 75% in high RH, while the proportion of dicarbonyls

in oligomer forms exhibited the opposite trend. Hydrates play a dominant role in dilute solutions under high RH conditions

with a relatively high aerosol liquid water concentration, which might hinder oligomer formation. And large quantities of

oligomers, including dimers and trimers, would form until the aerosol liquid concentration became greater than 1 M (Liggio

et al., 2005b) when RH decreased. However, the product distribution here was simulated based on the bulk-phase mechanisms

and higher ionic strength in aerosol phase would influence reaction equilibria and rate constant (Ervens and Volkamer, 2010;

Mcneill, 2015). The lack of quantitative reaction rate in aerosol phase could contribute more uncertainties to the simulation,

whereas, the RH dependence of product distribution and the order of magnitude of estimated $K_p$ values were close to those in

aerosol-phase and the roughly simulation could help to understand the reversible partitioning pathways of dicarbonyls.

Combined with the vapor pressure of dominant products published in previous studies (Hastings et al., 2005; Axson et al.,

2010), their gas-particle partitioning coefficient can be roughly estimated and can effectively fit the field-measured values, as

shown in Fig. 2c. The estimated gas-particle partitioning coefficients in this study are five orders of magnitude higher than the

theoretical ones but still 1–2 orders of magnitude lower than the field-measured coefficients, especially in winter. The

difference between the estimated partitioning coefficients and the field-measured ones suggests that the current understanding

of the equilibrium in reversible formations cannot reasonably explain the gas-particle partitioning processes of dicarbonyls.

There still exist extra pathways of reversible formation. Cross-oligomerization of glyoxal and methylglyoxal is nonnegligible

and could form similar molecular structure products and contribute to SOA yield (Schwier et al., 2010). Esterification and

amination of diols also occur in aerosol liquid water but are negligible compared to hydration and polymerization (Zhao et al.,

2006). However, these reactions are not further discussed here. The hydrates and oligomers mentioned above are the dominant

forms of glyoxal/methylglyoxal in the particle phase, while the higher molecular oligomers up to nonamer could also exist

with a relatively smaller but still significant fraction at equilibrium. Although the reactions are thermodynamically reversible,

upon evaporation of the aerosol liquid water, the oligomer formation is faster than the evaporation of dehydrated dicarbonyls,

and the dicarbonyl evaporation is limited (Liggio et al., 2005b; Loeffler et al., 2006). This results in relatively stable oligomers

and yielding SOA. Moreover, other nucleophilic species may also form oligomers with glyoxal and methylglyoxal and

effectively prevent their evaporation. Besides reversible pathways, higher carbon number products with lower volatility were mainly formed through irreversible pathways, such as radical reactions (e.g., OH radicals), which are fully discussed in the next section.

**3.3 Irreversible pathways**

**3.3.1 Irreversible pathways driven by hydroxyl radicals**

Reactive uptake driven by hydroxyl radicals (OH) is the dominant process for glyoxal and methylglyoxal in their irreversible gas-particle partitioning pathways. Compared to other irreversible pathways, such as imidazole formation, glyoxal/methylglyoxal + OH chemistry occurs on much shorter timescales (Teich et al., 2016). The reaction is the initial step for most radical-based chemistry of glyoxal/methylglyoxal and has been proven to be an important source of SOA in both cloud/fog droplets and wet aerosols (Tan et al., 2012; Lim et al., 2013), producing low-volatility products such as organic acids, large multifunctional humic-like substances, and oligomers. The proposed mechanism for the irreversible pathway of glyoxal and methylglyoxal driven by hydroxyl radicals in aerosols is shown in Fig. S7. The OH radicals in aerosol liquid water are mainly from the direct uptake of gas-phase OH radicals with a Henry's law constant of 30 M/atm (Faust and Allen, 1993) and Fenton reactions, and Fenton reactions are closely related to hydrogen peroxide, iron ions, and manganese ions in the particle phase. The sources of OH radicals are one of the major uncertainties in SOA formation (Ervens et al., 2014).

The calculated $\gamma$ and $k_{eff, uptake}$ values for different seasons are listed in Table 3. The reactive uptake coefficients of glyoxal were in the range $10^{-4}$–$10^{-2}$, and the average value of $8.0 \times 10^{-3}$ in this study was close to the ones representing the loss of glyoxal by surface uptake during the KORUS-AQ campaign in a very recent studies (Kim et al., 2022). And the value slightly exceeded the one commonly used in model simulations ($\gamma = 2.9 \times 10^{-3}$), which was based on an experimental study for $(NH_4)_2SO_4$ aerosols at 55% RH (Liggio et al., 2005a), and also far outweighs the uptake coefficients of glyoxal on clean and acidic gas-aged mineral particles ($\gamma = 10^{-6}$–$10^{-4}$) (Shen et al., 2016), implying that a real atmospheric aerosol provides a far more reactive interface for physiochemical processes than that of mineral particles. Moreover, uptake coefficients for methylglyoxal were with an average value of $2.0 \times 10^{-3}$ and were higher than those reported in other experimental studies, which varied from $10^{-6}$ to $10^{-3}$ (Curry et al., 2018; De Haan et al., 2018). On the one hand, conflicting with previous experimental results (Waxman et al., 2015), methylglyoxal exhibited an unexpected salting-in effect in real atmosphere due to much more complex compositions and higher ionic strength in ambient particles, which was also reported in other observational studies (Shen et al., 2018; Cui et al., 2021). And the higher Henry's law coefficient values in Eq.4 could lead to higher uptake coefficient values. One the other hand, a recent study also provided direct experimental evidence to confirm that methylglyoxal is more reactive and have larger uptake coefficients on seed particles under atmospherically relevant concentrations (Li et al., 2021).The $\gamma$ values for both glyoxal and methylglyoxal exhibited similar seasonal variations, which were lowest in summer and reached their highest

in winter. This seasonal variation could be attributed to RH variation and particle composition. Moreover, the effective uptake rate ($k_{eff, uptake}$), which is regarded as a pseudo-first-order reaction rate, is a net result of competition between reversible and irreversible processes, and it varied from $10^{-4}$ s$^{-1}$ to $10^{-5}$ s$^{-1}$ in the real atmosphere in this study. As shown in Fig. 3a, the negative dependence of $k_{eff, uptake}$ on RH also confirmed that the irreversible uptake of dicarbonyls could be inhibited in high RH conditions. What's more, as we can see in Figure 3b, the irreversible uptake increased exponentially with increasing SNA (SNA: Sulfate, Nitrate and Ammonia) concentrations, mainly because that higher SNA concentrations always occurred in lower RH conditions with lower aerosol liquid water content (Figure S8) and the irreversible uptake was promoted by combined efforts of RH effects and ion effects. Whereas, for a given RH, uptake coefficients γ for both glyoxal and methylglyoxal showed a weak dependence on the ratio of SNA (Sulfate:Ammonia and Sulfate:Nitrate) with significant scatter (Fig. S9).

Moreover, it was worth noting that under extremely low RH (<40%), the aerosol was not completely deliquescent, and the uptake coefficients based on Henry's law could not explain the irreversible pathways. Previous research indicated that the irreversible uptake of dicarbonyls could still occur under a low RH condition (Liggio et al., 2005a; De Haan et al., 2018), and that these uptake values were generally lower due to the inefficient reactive uptake process onto the crystallized aerosols.

### 3.3.2 Reactive uptake of dicarbonyl compounds

We could not directly measure the particulate concentration of dicarbonyls via an irreversible pathway, as the dicarbonyls irreversibly reacted with oxidative radicals on aerosols. To quantitively evaluate the contribution of the irreversible pathway of dicarbonyls, we calculated their average concentration based on Eqs. (S3)–(S7) in the Supplement with the calculated γ values in this study. The samples estimated here were collected under high RH conditions (RH > 40%) because of the calculation limitation of irreversible uptake coefficients. Although the products of irreversible pathways could not be directly detected in particle phase and didn't directly contribute to the increase of particulate dicarbonyls, the irreversible pathways could contribute to the decrease of gasous dicarbonyls and well explain the overestimation of modeled dicarbonyl mixing ratios, which is about 3-6 times higher than the observed ones (Volkamer et al., 2007; Ling et al., 2020).

The total particulate concentration of glyoxal and methylglyoxal via irreversible pathway varied from several to more than 100 nanograms per microgram PM$_{2.5}$ (ng/μg PM$_{2.5}$), and it was strongly dependent on RH, as shown in Fig. 3c, which generally decreased with increasing RH. Concentrated inorganic solutions and relatively higher ionic strength in aerosol water under low RH conditions could jointly contribute to the hydration of dicarbonyls, the products of which could easily participate into the following irreversible radical reactions via H-abstraction.

To further discuss the product distribution of the reaction of glyoxal/methylglyoxal with hydroxyl radicals, we used the kinetic mechanisms of glyoxal/methylglyoxal + OH chemistry proposed by Lim et al.(2013) on the basis of a 0-D box model with a

steady-state approach. The average OH radical concentration setting in the modeling was $3.2 \times 10^{-12}$ M, which is based on the

hypothesis of the Henry equilibrium of OH radicals between the gas and particle phase (Sander, 2015; Shen et al., 2018).

Oxalate can be considered as a tracer for this aqueous chemistry, since it does not have any other significant chemical sources.

Oxalate was detected in the particle-phase samples by ion chromatography. The modeling results of oxalate concentration

agreed well with the measured values, and their deviations were in the considered range (Fig. S10). Meanwhile, we can

estimate the distribution of major products in irreversible glyoxal/methylglyoxal-OH radical chemistry under different RH

conditions, as illustrated in Figure 3d. Generally, oxalate is the major product in wet aerosols, contributing ~60%, and its

proportion increases with increasing RH. Besides oxalate, oligomers also play significant roles in glyoxal/methylglyoxal-OH

radical chemistry with a contribution of ~30%, and their proportion is maximum under relatively low RH conditions. The RH

dependence of the product distribution could mainly be attributed to the particulate concentration of glyoxal/methylglyoxal,

which significantly affects the OH radical chemistry. With relatively high carbonyl concentrations (0.1–10 M) in aerosol liquid

water, self-reactions of organic molecules become more favorable, resulting in new carbon–carbon bonds and high molecular

weight oligomers via radical–radical chemistry (Lim et al., 2013). Moreover, besides OH radical chemistry, reaction with

sulfate and ammonium also contribute to the oligomer formation and irreversible uptake of gaseous dicarbonyls (Ortiz-

Montalvo et al., 2014; Lin et al., 2015; Lim et al., 2016). The oligomer proportion could be more than 30% in concentrated

carbonyl solutions (~0.1 M) and only account for 1% in diluted solutions (~0.01 M).

**3.4 Relative importance of two partitioning pathways**

Table 4 summarizes the particulate concentration of glyoxal and methylglyoxal via reversible and irreversible pathways in

different seasons. The average particulate concentrations of glyoxal (0.43 ng/μg in the reversible pathway and 24.26 ng/μg in

the irreversible pathway) were generally higher than those of methylglyoxal (0.25 ng/μg in the reversible pathway and 16.53

409  ng/μg in the irreversible pathway), mainly due to the relatively higher water solubility and reactivity of glyoxal. Comparing

two gas-particle partitioning processes, the irreversible pathway played extremely dominant roles and generally accounted for

96.7% and 95.0% for glyoxal and methylglyoxal, respectively. The proportion of the irreversible pathway varied from 90% to

99.9% and reached its highest in summer for glyoxal (98.8%) and in autumn for methylglyoxal (99.2%), while it was minimum

in winter (92.9% for glyoxal and 92.8% for methylglyoxal). Overall, the irreversible pathway played a dominant role in the

gas-particle partitioning process for both glyoxal and methylglyoxal in the real atmosphere, while the reversible pathway was

also substantial and nonnegligible, especially in winter, with an proportion of ~10%. Furthermore, as discussed above, the

particulate concentrations of dicarbonyls and their relative importance were influenced by environmental factors such as

relative humidity and particle composition, which could jointly influence both the reversible and irreversible pathways of

dicarbonyls. As shown in Figure 4, the proportion of irreversible pathways in the gas-particle partitioning process for

dicarbonyls increased with aqueous SNA concentrations, and reached maxium when SNA concentrations were more than 100

M under low RH conditions. Moreover, higher organic concentrations in aerosol may lead to an OH-limit environment,

hindering the irreversible pathways driven by radicals and influencing the relative importance of the two pathways (Waxman

et al., 2013; Ervens et al., 2014). But the OH limitations are still enclusive due to the uncertainties in the sources of OH in

aerosol particles(Herrmann et al., 2010).

Comprehensively considering the contribution of both reversible pathways and irreversible pathways occurred in gas-particle

partitioning processes could benefit the ambient dicarbonyls simulations. Ling et al. (2020) found that the observation and

simulation of the gas-phase concentration level of dicarbonyls could reach reasonable agreement when the irreversible uptake

and reversible partitioning were incorporated into the model, as these jointly contribute ~62% to the sink of dicarbonyls.

Moreover, the contribution of gas-particle partitioning processes of dicarbonyls to SOA formation were higher as the two

partitioning pathways were jointly considered. In this study, gas-particle partitioning processes of dicarbonyls accounted for a

relatively large proportion of total particle mass ($PM_{2.5}$), on the average of ~5% considering both reversible and irreversible

gas-particle partitioning pathways. Since a large fraction of $PM_{2.5}$ mass in Beijing consists of SOAs (~30%) (Huang et al.,

2014), we could roughly estimate the contribution of gas-particle partitioning processes of dicarbonyls to SOA yields (by

mass). There were approximately 25% SOAs formed from glyoxal and methylglyoxal in this study. However, the particulate

dicarbonyls calculated here only contained simple reversible pathways and irreversible pathways driven by OH radicals. More

complicated chemical processes, such as $NO_3$ radical chemistry, were not considered, which still resulted in the

underestimation of their contribution to SOA formation.

**4 Conclusions**

We simultaneously measured glyoxal and methylglyoxal concentration in the gas and particle phase in different seasons over

urban Beijing. Based on field-measured data, the field-derived gas-particle partitioning coefficients were calculated and found

to be 5–7 magnitudes higher than the theoretical values. Such a large discrepancy provides field evidence that the gas-particle

partitioning process does not occur by physical absorption alone but also results from the combined and simultaneous effects

of reversible and irreversible pathways. Hydration and oligomerization occurred in the reversible pathway, producing

compounds with lower volatility in the condensed phase, and the irreversible pathway could accelerate the uptake of gaseous

dicarbonyls. The two pathways jointly contributed to the underestimation of gas-particle partitioning of dicarbonyls.

This study systemically considers both reversible and irreversible pathways in the ambient atmosphere for the first time.

Compared to the reversible pathways, the irreversible pathways play a dominant role in the gas-particle partitioning process

for dicarbonyls, accounting for ~90% of this process. We recommend the irreversible reactive uptake coefficient for glyoxal

and methylglyoxal in different seasons in the real atmosphere. The values we calculated here are higher than those used in model simulations to date, especially for methylglyoxal which exhibits an unexpected salting-in effect under an atmospheric-relevant concentration. We expect the application of these parameterizations will increase the calculated contribution of irreversible uptake of dicarbonyls to SOA formation and narrow the gap between model predictions and field measurements of ambient dicarbonyl concentrations. Moreover, relative humidity and inorganic particle compositions are defined as the most important factor influencing particulate concentration and product distribution of dicarbonyls via both reversible and irreversible pathways, implying the significance of considering different RH conditions in dicarbonyl SOA simulations.

Furthermore, we note that there may be other potential explanations for the increase in particle mass caused by dicarbonyls and the uncertainty in the gas-particle partitioning process, including physical adsorption, reversible pathways and irreversible pathways. Physical adsorption of dicarbonyls could be enhanced by water-soluble organics and mineral dust. Other reversible pathways, such as adducts formed from glyoxal with inorganic species, could also promote the gas-particle partitioning process. Irreversible pathways driven by other oxidants, such as $NO_3$ radicals, can also perform a substantial role. Shen et al. (2016) found that glyoxal could irreversibly produce formic acid, glycolic acid, and oligomers on particles without illumination or extra oxidants. Besides gas-particle partitioning, particulate dicarbonyls formed via the heterogeneous reaction of VOCs could contribute to the uncertainty in partitioning research. Dong et al. (2021) recently revealed that aqueous photooxidation of toluene could yield glyoxal and methylglyoxal via a ring-cleavage process. Overall, the real gas-particle partitioning process of glyoxal and methylglyoxal is more complicated and their contribution to SOA formation is still indistinct; thus, more laboratory experiments and field measurements are urgently needed to improve our understanding of the gas-particle partitioning process for glyoxal and methylglyoxal.

*Data availability.* The data are accessible by contacting the corresponding author (zmchen@pku.edu.cn).

*Author contributions*. In the framework of the five field measurements in different seasons, ZC and JH designed the study, and JH performed all carbonyl measurements used in this study, analyzed the data, and wrote the paper. ZC helped interpret the results, guided the writing, and modified the manuscript. XQ and PD contributed to the methods of sampling and analyzing gas- and particle-phase carbonyls. All authors discussed the results and contributed to the final paper.

*Competing interests*. The authors declare that they have no conflict of interest.

*Acknowledgements*. This work was funded by the National Natural Science Foundation of China (Grant number 41975163). We also thanked Shiyi Chen at Peking University for the providing the data for the meteorological parameters, trace gases and

PM$_{2.5}$ mass concentrations.

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

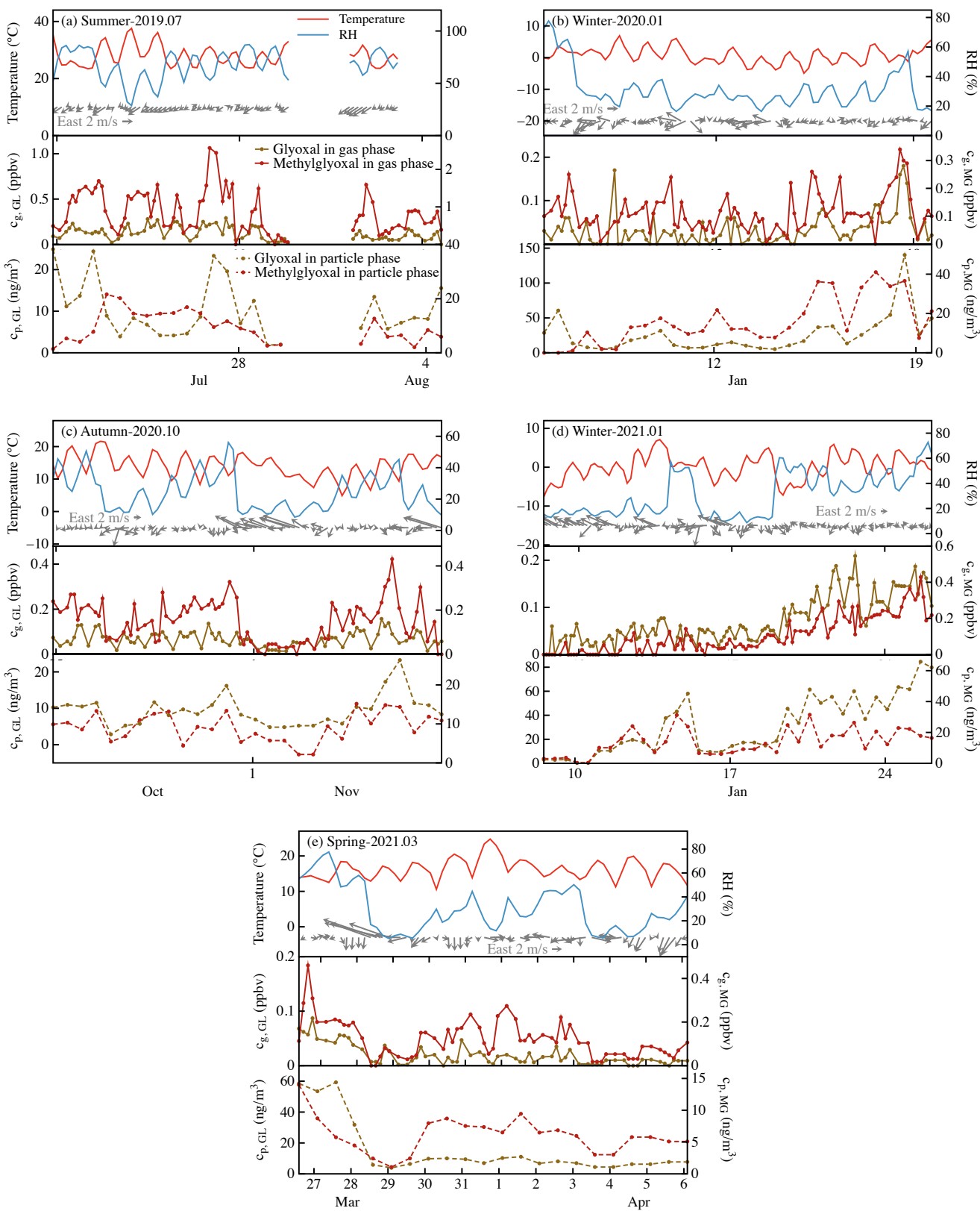

Figure 1: Time series of meteorological parameters and gas- and particle-phase glyoxal and methylglyoxal observed in different

seasons: (a) summer, 2019.07.20-2019.08.04; (b) winter, 2020.01.05-2020.01.19; (c) autumn, 2020.10.24-2020.11.07; (d)

winter, 2021.01.08-2021.01.26; (e) spring, 2021.03.26-2021.04.06.

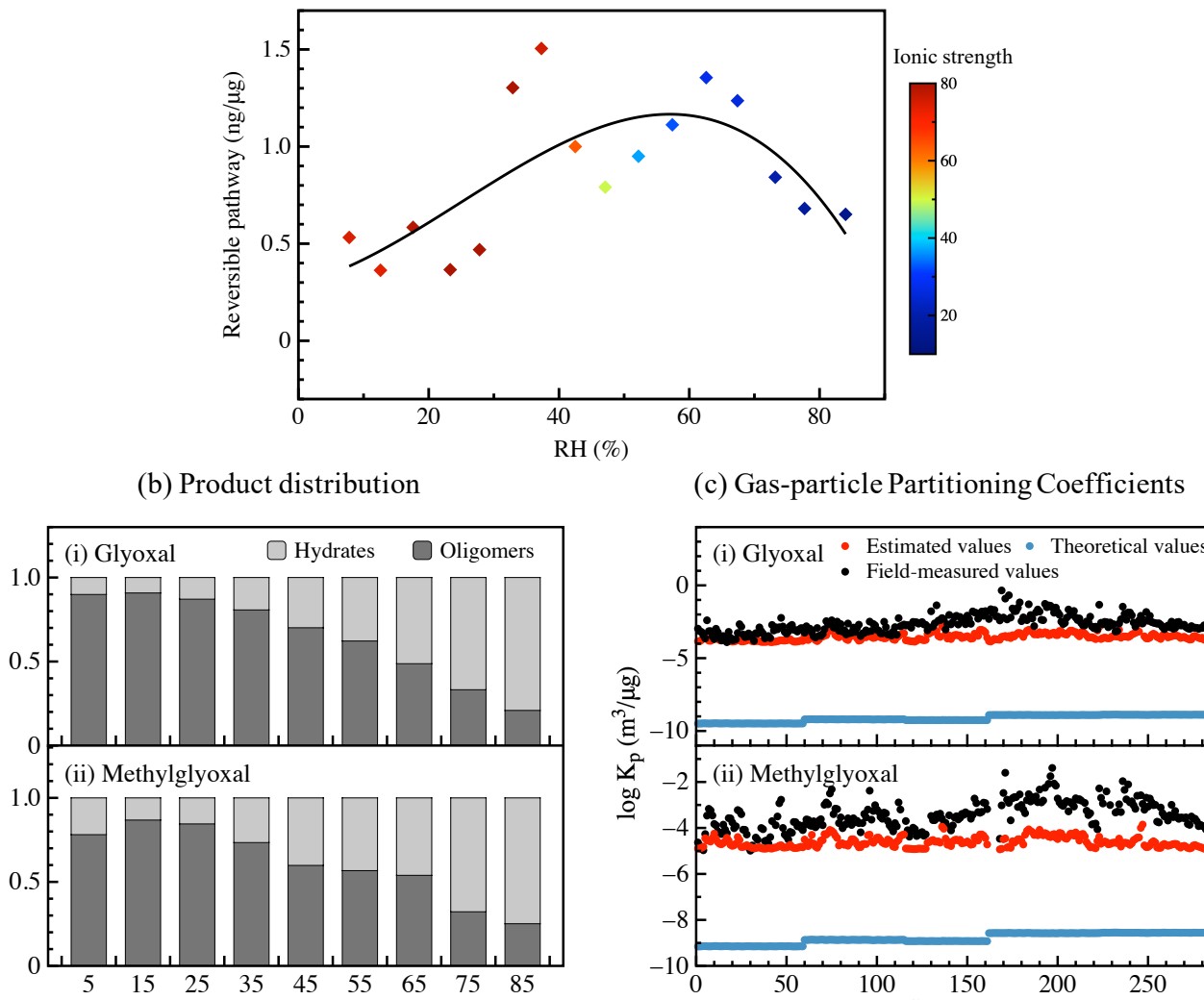

Figure 2: Gas-particle partitioning of dicarbonyls via reversible pathways. (a) The RH dependence of particulate concentrations of dicarbonyl via reversible pathways. (b) The product distribution for (i) glyoxal and (ii) methylglyoxal under different RH conditions. (c) The gas-particle partitioning coefficients for (i) glyoxal and (ii) methylglyoxal. The black, red, and blue circles refer to field-measured values, estimated values by the proposed mechanism, and theoretical values calculated by Pankow's absorptive model, respectively.

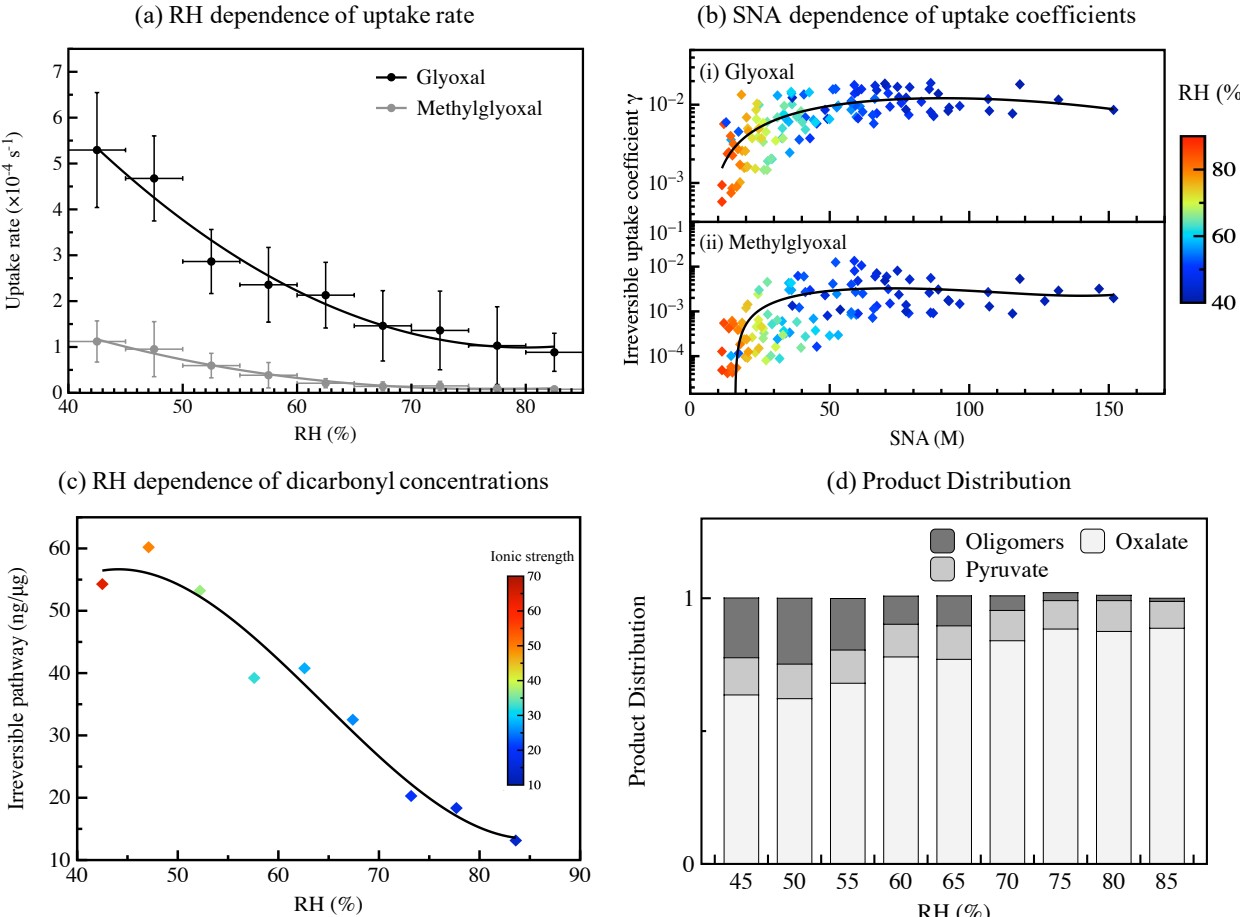

Figure 3: Gas-particle partitioning of dicarbonyls via irreversible pathways. (a) The RH dependence of irreversible uptake rate

for glyoxal and methylglyoxal. (b) The SNA dependence of uptake coefficients for (i) glyoxal and (ii) methylglyoxal, SNA

refers to the concentration of sulfate, nitrate and ammonia in wet aerosols. (c) The RH dependence of particulate concentrations

of dicarbonyl via irreversible pathways. (d) The corresponding modeled product distribution in wet aerosols under different

RH conditions.

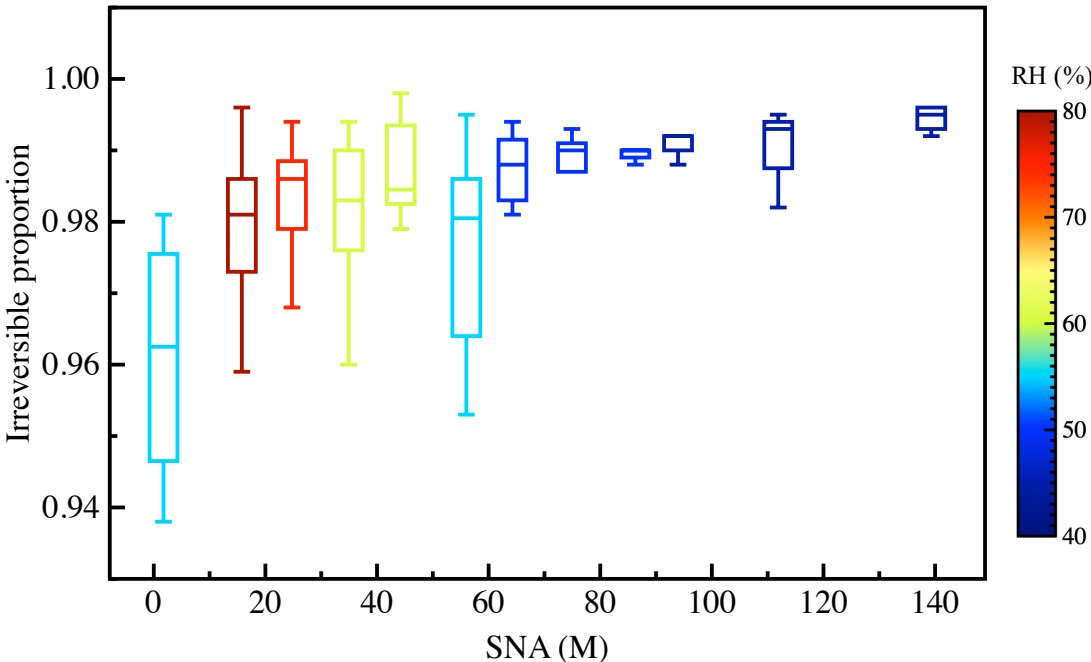

Figure 4: Correlation between the proportion of the irreversible pathway in gas-particle partitioning process for dicarbonyls

and aqueous sulfate, nitrate, and ammonia (SNA) concentration in ambient aerosols under different relative humidity

conditions.

Table 1: Statistical data for the α-dicarbonyls in gas and particle phase in different seasons.

| Season | Dates of the measurements | Gas phase (ppbv) | | Particle phase (ng m$^{-3}$) | |
|---|---|---|---|---|---|
| | | Glyoxal | Methylglyoxal | Glyoxal | Methylglyoxal |
| summer | 2019.07.20-08.04 | $0.13 \pm 0.07$ | $0.87 \pm 0.54$ | $10.18 \pm 6.63$ | $9.50 \pm 5.62$ |
| spring | 2021.03.26-04.06 | $0.02 \pm 0.02$ | $0.12 \pm 0.08$ | $15.24 \pm 17.50$ | $6.07 \pm 2.79$ |
| autumn | 2020.10.24-11.07 | $0.07 \pm 0.03$ | $0.15 \pm 0.09$ | $9.33 \pm 4.24$ | $9.15 \pm 3.62$ |
| winter | 2020.01.05-01.19, 2021.01.08-01.26 | $0.06 \pm 0.05$ | $0.11 \pm 0.09$ | $28.77 \pm 25.33$ | $14.61 \pm 10.15$ |

Table 2: Comparison of the field-measured partitioning coefficient $K^f$ values for the dicarbonyls and their corresponding theoretical $K^t$ values in different seasons.

| Coefficients | Dicarbonyl | Season | $K^f$ | | $^a K^t$ | $K^f / K^t$ |
|---|---|---|---|---|---|---|
| | | | Average | Range | | |
| Gas-particle partitioning coefficients ($m^3 \cdot \mu g^{-1}$) | Glyoxal | summer | $8.11 \times 10^{-4}$ | $(1.25\text{-}58.6) \times 10^{-4}$ | $3.27 \times 10^{-10}$ | $2.48 \times 10^{6}$ |
| | | autumn | $2.14 \times 10^{-3}$ | $(2.61\text{-}224) \times 10^{-4}$ | $6.27 \times 10^{-10}$ | $3.41 \times 10^{6}$ |
| | | spring | $1.43 \times 10^{-2}$ | $(0.08\text{-}14.6) \times 10^{-2}$ | $5.59 \times 10^{-10}$ | $3.55 \times 10^{7}$ |
| | | winter | $1.30 \times 10^{-2}$ | $(0.067\text{-}44.2) \times 10^{-2}$ | $1.27 \times 10^{-9}$ | $1.02 \times 10^{7}$ |
| | Methylglyoxal | summer | $1.49 \times 10^{-4}$ | $(0.833\text{-}169) \times 10^{-5}$ | $7.10 \times 10^{-10}$ | $2.10 \times 10^{5}$ |
| | | autumn | $9.55 \times 10^{-4}$ | $(0.65\text{-}86.9) \times 10^{-4}$ | $1.35 \times 10^{-9}$ | $7.07 \times 10^{5}$ |
| | | spring | $1.06 \times 10^{-3}$ | $(0.42\text{-}108) \times 10^{-4}$ | $1.21 \times 10^{-9}$ | $8.77 \times 10^{5}$ |
| | | winter | $2.60 \times 10^{-3}$ | $(0.34\text{-}410) \times 10^{-4}$ | $2.72 \times 10^{-9}$ | $9.93 \times 10^{5}$ |
| Henry's law coefficients ($M \cdot atm^{-1}$) | Glyoxal | summer | $1.96 \times 10^{8}$ | $(1.71\text{-}167) \times 10^{7}$ | $3.29 \times 10^{5}$ | $6.11 \times 10^{2}$ |
| | | autumn | $5.08 \times 10^{8}$ | $(1.88\text{-}10.2) \times 10^{8}$ | $1.14 \times 10^{6}$ | $3.63 \times 10^{3}$ |
| | | spring | $2.53 \times 10^{9}$ | $(1.23\text{-}139) \times 10^{8}$ | $9.03 \times 10^{5}$ | $1.92 \times 10^{4}$ |
| | | winter | $1.04 \times 10^{9}$ | $(1.37\text{-}55.4) \times 10^{8}$ | $4.15 \times 10^{6}$ | $2.55 \times 10^{3}$ |
| | Methylglyoxal | summer | $4.92 \times 10^{7}$ | $(1.70\text{-}363) \times 10^{6}$ | $2.73 \times 10^{3}$ | $1.88 \times 10^{4}$ |
| | | autumn | $8.52 \times 10^{7}$ | $(3.66\text{-}15.7) \times 10^{7}$ | $9.50 \times 10^{3}$ | $2.00 \times 10^{5}$ |
| | | spring | $1.33 \times 10^{8}$ | $(5.22\text{-}456) \times 10^{6}$ | $7.49 \times 10^{3}$ | $1.36 \times 10^{5}$ |
| | | winter | $2.63 \times 10^{8}$ | $(1.03\text{-}175) \times 10^{7}$ | $3.44 \times 10^{4}$ | $9.01 \times 10^{4}$ |

[a] Theoretical gas-particle partitioning coefficients were calculated on the basis of Eq. (3) and theoretical Henry's law coefficients here referred to the Henry's law constant in pure water, which were calculated on the basis of Eq.(S1)-(S2) (Ip et al., 2009; Sander, 2015).

Table 3: Summary of calculated uptake coefficients γ and effective uptake rate coefficient $k_{eff,\ uptake}$ in different seasons for

glyoxal and methylglyoxal.

| Dicarbonyl | Season | T (K) | RH (%) | γ (×10$^{-3}$) | | | $k_{eff,\ uptake}$ (s$^{-1}$) (×10$^{-4}$) |
|---|---|---|---|---|---|---|---|
| | | | | Average | Min | Max | |
| Glyoxal | Summer | 301.1 | 67.7 | 4.15 | 0.12 | 7.30 | 1.61 |
| | Autumn | 287.2 | 45.4 | 8.62 | 0.29 | 12.9 | 4.83 |
| | Spring | 289.4 | 54.0 | 11.7 | 1.24 | 14.9 | 6.85 |
| | Winter | 273.5 | 54.0 | 10.6 | 0.56 | 14.4 | 3.59 |
| | [a] **General** | **287.8** | **59.0** | **8.0** | **0.46** | **11.4** | **3.38** |
| Methylglyoxal | Summer | 301.1 | 67.7 | 1.01 | 0.02 | 2.09 | 0.25 |
| | Autumn | 287.2 | 45.4 | 1.83 | 0.04 | 3.94 | 0.92 |
| | Spring | 289.4 | 54.0 | 2.36 | 0.06 | 4.34 | 0.69 |
| | Winter | 273.5 | 54.0 | 3.45 | 0.11 | 5.83 | 0.77 |
| | [a] **General** | **287.8** | **59.0** | **2.0** | **0.05** | **3.8** | **0.55** |

[a] General is the average value of all the samples observed in the five field observations.

Table 4: Calculated relative importance of reversible and irreversible pathways in the gas-particle partitioning processes and

their contribution to the particulate matter.

| Season | Glyoxal | | Methylglyoxal | | Contribution to particulate matters |
|---|---|---|---|---|---|
| | [a] $[X]_{P, rever}$ | [b] $[X]_{P, irrever}$ | [a] $[X]_{P, rever}$ | [b] $[X]_{P, irrever}$ | |
| Summer | 0.17 (1.2%) | 18.87 (98.8%) | 0.25 (5.5%) | 20.55 (94.5%) | 3.98% |
| Autumn | 0.14 (0.7%) | 23.91(99.3%) | 0.12 (0.8%) | 17.02 (99.2%) | 4.12% |
| Spring | 0.26 (1.9%) | 15.94 (98.1%) | 0.09 (1.5%) | 14.16 (98.5%) | 3.05% |
| Winter | 0.89 (7.1%) | 34.70 (92.9%) | 0.38 (7.2%) | 12.59 (92.8%) | 4.86% |
| [c] **General** | **0.43 (3.3%)** | **24.26 (96.7%)** | **0.25 (5.0%)** | **16.53 (95.0%)** | **4.15%** |

[a] $[X]_{P, rever}$ is the concentration of particle-phase carbonyl via reversible pathway (ng·µg$^{-1}$) and its proportion (%).

[b] $[X]_{P, irrever}$ is the concentration of particle-phase carbonyl via irreversible pathway (ng·µg$^{-1}$) and its proportion (%).

[c] General is the average value of all the samples observed in the five field observations.