# Peer review of "Reversible and irreversible gas-particle partitioning of dicarbonyl compounds observed in the real atmosphere."

_Atmospheric Chemistry and Physics, 2022_

## Author Comment (AC1)

**Response to Reviewer #1**

We gratefully thank you for your constructive comments and thorough review. Below are our point-by-point responses to your comments.

(Q=Question, A=Answer, C=Change in the revised manuscript)

**General Comments:**

Hu and coauthors describe a set of field experiments designed to investigate the gas/particle partitioning of glyoxal and methylglyoxal in Beijing as a function of season. Gas phase dicarbonyls were collected using DNPH-doped cartridges and particle phase dicarbonyls were collected using a filter assembly. Both reversible and irreversible uptake pathways are considered using supporting measurement data and irreversible pathways are found to be dominant for both dicarbonyls in all seasons, although reversible uptake (self-reaction, oligomerization) becomes more relevant in the winter. As expected, the field data demonstrate particle phase concentrations that are orders of magnitude higher than those expected based solely on absorptive partitioning theory. This study, however, is particularly useful in demonstrating the estimated dominance of the irreversible pathway in all seasons, and in presenting real-world SOA contributions for these dicarbonyls at this urban location (approximately 25% of Beijing SOA is assigned to glyoxal/methylglyoxal uptake processes). Overall, I find the manuscript to be well written with comprehensive consideration of the relative importance of reversible and irreversible uptake pathways and their impact on SOA production. These data should be useful for optimizing dicarbonyl uptake in SOA modeling efforts.

A: We highly appreciate your comments and suggestions. The questions you mentioned are answered as follows.

**Major Comments:**

Q1: Line 86: How were positive artefacts from direct deposition of gas phase glyoxal/methylglyoxal on the filter surfaces accounted for? This could bias the partitioning result from Equations 1 and 3 and should be discussed in the text.

A1: Thanks for your suggestion. Collecting particulate glyoxal/methylglyoxal by quartz filter without denuders might lead to positive artifacts as gaseous dicarbonyls could be adsorbed to the filter (Hart and Pankow, 1994; Mader and Pankow, 2001;

Liggio, 2004). In field observations, Odabasi and Seyfioglu (2005) revealed that about 36% of the measured particulate formaldehyde was caused by adsorbed formaldehyde on quartz filters, and Shen et al. (2018) found that the fractions of gaseous glyoxal/methylglyoxal in particulate samples were usually lower than 20%. To evaluate the possible adsorption artifacts, throughout our field observations, we placed a backup quartz filter after the particle sampling quartz filter using an independent filter holder. The first filters would collect the particles and adsorbed gaseous dicarbonyls, while the second filter would only collect gaseous carbonyls. And the ratio of measured dicarbonyls in second filter to that in the first were lower than 20%. The particulate concentrations of dicarbonyls used in this study were already corrected by the possible adsorption artifacts. And the partitioning results from Equations 1 and 3 were all calculated by calibrated concentrations of dicarbonyls and the artifacts caused by filter adsorption would not change our conclusions. We have expanded the methods of field sampling in the revised manuscript.

C1: Lines 105-110 in Sect. 2.1:

To estimate the positive artifacts by adsorption of gas-phase dicarbonyls onto the filter (Hart and Pankow, 1994; Mader and Pankow, 2001; Liggio, 2004; Odabasi and Seyfioglu, 2005), throughout our previous field observations, we placed a backup quartz filter after the particle sampling quartz filter using an independent filter holder. The sampling filters would collect the particles and adsorbed gaseous dicarbonyls, while the backup filter would only collect gaseous dicarbonyls. And the ratio of measured dicarbonyls in second filter to that in the first were lower than 20%, which was equal to the previous study (Shen et al., 2018). And the particulate concentrations of dicarbonyls used in this study were already corrected by the possible adsorption artifacts.

Q2: I think the manuscript would benefit from an expanded discussion of assumptions used to calculate the reversible and irreversible pathways. Calculations of the irreversible pathway involve the particle phase and gas phase monomer dicarbonyl concentrations to be known and these data are derived from analysis of the extracts. But the particle phase monomer extract concentration will also include the contributions from the reversibly formed products present in the extract. Expanding the

discussion of deriving the cp term in the formulas and what exactly it represents would be useful for readers.

A2: Thanks for your suggestion. The  $c_p$  term in the formulas referred to the measured particulate carbonyls which was derived from the analysis of the extracts and it included the dissolved dicarbonyls monomers and their reversibly formed products (the reversible formed products could easily revert to their original monomer form during extraction and derivation). The concentrations of dissolved dicarbonyl monomers were estimated using theoretical Henry's law coefficients, which are used to determine the physical solubility of carbonyls (e.g.,  $K_H=5 \text{ M} \cdot \text{atm}^{-1}$  for glyoxal) (Schweitzer et al., 1998). The results were negligible compared to the concentrations of carbonyls in hydrate and oligomer forms. In section 3.2, we have calculated the product distribution of reversible reactions based on reversible reaction mechanisms, which could help to understand what exactly the cp term represented. The cp term included hydrates with proportion of 10%-50%, oligomers with proportion of 50%-90%, and monomers with proportion of ~1%. We have expanded the discussion of deriving  $c_p$  term in the formulas in the revised manuscript.

C2: Lines 170-172 in Sect. 2.3:

 $C_p$  ( $\mu g \cdot m^{-3}$ ) is the concentrations of dicarbonyls in the particle phase, which is derived from the analysis of extracts, including monomers and their reversibly formed products (the product distribution is discussed in Section 3.2).

**Minor and Technical Comments:**

Q3: Figure 2c: Change the color of one of the grey traces

A3: We have changed the color of the traces in Figure 2c as follows.

C3:

Figure 2c: The gas-particle partitioning coefficients for (i) glyoxal and (ii) methylglyoxal. The black, red, and blue circles refer to field-measured values, estimated values by the proposed mechanism, and theoretical values calculated by Pankow's absorptive model, respectively.

Q4: Figure 3: Define SNA in caption

**A4: We have defined SNA in caption in Figure 3.**

Q5: Figures throughout: Colorscales should go from lower values in blue to higher values in red to be consistent with general uses in the literature. It is counterintuitive for the reader for these to be the other way around. Also the axis scale in the colorscale legends has numbers increasing from right to left which is also confusing.

A5: Thanks for your suggestion. We have revised the color scales in all figures, which now consistently go from lower values in blue and to higher values in red. And we also revised the axis scale in the legends of color scale, the numbers of which now increase from bottom to top. The revised figures are presented as follows (take Figure 4 for example):

C5:

Figure 4: Correlation between the proportion of the irreversible pathway in gas-particle partitioning process for dicarbonyls and aqueous sulfate, nitrate, and ammonia (SNA) concentration in ambient aerosols under different relative humidity conditions.

Q6: Abstract, change last line to present tense. eg "To our knowledge, this article is the first to…"

A6: Thanks for your suggestion. We have changed last line to present tense.

C6: Lines 22-25 in Abstract:

To our knowledge, this study is the first to systemically examine both reversible and irreversible pathways in the ambient atmosphere, strives to narrow the gap between model simulations and field-measured gas-particle partitioning coefficients, and reveals the importance of gas-particle processes for dicarbonyls in SOA formation.

Q7: Line 33: reorder the two references

A7: Thanks for your suggestion. We have reordered the two references.

Q8: Line 36: consider rephrasing to "lost in the gas phase by photolysis, oxidation by OH radicals, and dry deposition" as OH oxidation is a photochemical reaction

A8: Thanks for your suggestion. We have rephrased the sentences as follows.

C8: Lines 40-41 in Sect.1:

Considering the atmospheric sink, glyoxal and methylglyoxal can be lost in the gas phase by self-photolysis, oxidation by active radicals (like OH radicals, NO3 radicals) and wet/dry deposition.

Q9: Line 40: "Although they have relatively high..."; Line 90: "common"; Line 94: "time resolution"; Line 167: "soluble"; Line 170: "close to"; Line 180: "real atmosphere"; Table 4: "Particulate matter"; Line 270: "close to"

A9: Thanks for your suggestion. We have revised that.

Q10: Line 41: How relevant is adsorption to surfaces vs absorption into the bulk particle phase material? Worth discussing here

A10: Thanks for your suggestion. We have discussed the relevance of adsorption to surfaces vs adsorption into the bulk particle phase material as follows.

C10: Lines 47-55 in Sect.1:

The surface-adsorbed dicarbonyls could alter the properties of the particle's surfaces and the organic surface films could act as a kinetic barrier to gas-aerosol mass transport and thereby influence particle equilibration and water/gas uptake (Donaldson and Vaida, 2006). Upon physical adsorption, besides desorption or reaction at the surface, dicarbonyls could undergo solvation and incorporation into the bulk liquid, and then they could go through diffusion and chemical reactions in the bulk phase. The product may return into surfaces and gas phase, or stay in the bulk phase (Paul et al., 2011). Moreover, chemical reactions occurred at the surface or in the bulk phase could in turn accelerate the physical adsorption and greatly contribute to the formation and growth of atmospheric particulate matter. Whereas, as it is difficult to distinguish the surface reactions and bulk reactions in field observations, we regard both of them as particlephase reactions in this study.

Q11: Line 206: Worth noting that the observed RH dependence for the reversible pathway is consistent with the Healy et al 2009 chamber study reference

A11: Thanks for your suggestion. We have added the reference as follows.

C11: Lines 290-293 in Sect. 3.2:

The results exhibited a similar pattern to a previous study, in which the partitioning of glyoxal and methylglyoxal gradually increased as RH increased to 40%, peaked sharply around 50%, and subsequently decreased as RH increased towards 80% (Healy et al., 2009).

Q12: Line 284: Define SNA

A12: We have defined SNA in the revised manuscript.

Q13: Line 285: rephrase

A13: Thanks for your suggestion. We have rephrased the sentences as follows.

C13: Lines 382-386 in Sect. 3.3.2:

The total particulate concentration of glyoxal and methylglyoxal via irreversible pathway varied from several to more than 100 nanograms per microgram  $PM_{2.5}$  (ng/µg  $PM_{2.5}$ ), and it was strongly dependent on RH, as shown in Fig. 3c, which generally decreased with increasing RH. Concentrated inorganic solutions and relatively higher ionic strength in aerosol water under low RH conditions (Fig. S8) could jointly contribute to the hydration of dicarbonyls, the products of which could easily participate into the following irreversible radical reactions via H-abstraction.

---

## Author Comment (AC2)

**Response to Reviewer #2**

We gratefully thank you for your constructive comments and thorough review. Below are our point-by-point responses to your comments.

(Q=Question, A=Answer, C=Change in the revised manuscript)

**General Comments:**

Jingcheng Hu and co-authors have measured the gas-particle partitioning of dicarbonyl compounds, especially glycol and methylglyoxal, at field sites in China. This is a highly relevant topic for the readership of ACP. As far as I can tell (being a computational chemist, not an experimentalist), the study is well carried out, and the manuscript is well written. I can thus recommend publication subject to some fairly minor revisions.

A: We highly appreciate your comments and suggestions. The questions you mentioned are answered as follows.

**Major Comments:**

Q1: The authors spend a lot of time pointing out that the measured partitioning is much higher than what they call the "theoretical" values - the latter seem to correspond to values obtained for the partitioning coefficient (or Henry's law constant) of pure molecular glyoxal and methylglyoxal. However, as evident from their own introduction section, it is already very well known that the partitioning of these compounds is driven mainly by various reactions. For example, hydration alone is well-known to increase the Henry's law coefficient of glyoxal by about five orders of magnitude (as discussed e.g. in Ip et al 2009, https://doi.org/10.1029/2008GL036212, or Kampf et al 2013 cited in the manuscript). The authors contribution to separating reversible and irreversible pathways is substantial and valuable - but just reporting that partitioning is much stronger than the "theoretical" values is not really novel (or even that interesting), and this aspect of the abstract and discussion should be toned down. For example, the speculation about "misidentification" or "discrepancies" around lines 175-180 is not really warranted: we already know mechanisms which can easily explain at least most of the observed deviations from the "theoretical" pure-compound values.

A1: Thanks for your suggestion. We have toned down the discussion of discrepancy between the field-measured partitioning coefficients and the theoretical ones in Section 3.1.2 in our revised manuscripts, which had been fully discussed in previous studies (Ip

et al., 2009; Kampf et al., 2013). And we would focus on and expand the discussions of reversible and irreversible pathways, which is relatively substantial and valuable.

Q2: Concerning the saturation vapour pressures discussed around line 190: are there any estimates of the relative saturation vapour pressures of the reversible vs irreversible products? Both are of course much lower than the saturation vapour pressures of the parent dicarbonyls (this is quite well-known and obvious), but how do the two product sets compare with each other? This would be a very interesting parameter to know in terms of evaluating the atmospheric impact of the "reversible vs irreversible" competition.

A2: Thanks for your suggestion. We have expanded the discussion of saturation vapor pressures of the two product sets in our revised manuscript as follows.

C2: Lines 257-262 in Section 3.1.2:

Take glyoxal for example, the effective saturation vapor pressures of the product set in reversible pathways are ~$10^{-5}$ Torr in the real atmosphere (Shen et al., 2018). And the products of the irreversible pathways had much lower vapor pressure values than those of reversible pathways, for example, the vapor pressure of oxalic acids and ammonium oxalates are ~$10^{-5}$ Torr (Saxena and Hildemann, 1996) and $5.18\times 10^{-8}$ Torr (EPA, 2011), respectively, and those of glyoxal trimer dihydrates are ~$10^{-11}$ Torr at 20 °C (SPARC, 2003), indicating the irreversible pathways make larger contributions to the underestimation of partitioning processes of dicarbonyls.

Q3: Line 208: "strong and positive dependence on particle acidity (pH)". Please be clear here: did the concentration increase with acidity (i.e. with decreasing pH), or did it increase with pH? These are opposite things.

A3: Thanks for your suggestion. In this study, the particulate concentration of dicarbonyls via reversible pathways had a strong dependence on particle acidity under high RH conditions, which increased with increasing pH values (decreasing aerosol acidity) and reached highest when pH was close to 4. Whereas, as the other reviewers suggested, there is no gas-phase measurements to constrain the partitioning of the semi-volatile gases, which can lead to large deviations in the calculated pH by the thermodynamic model ISORROPIA-II from real world observations. Thus, we would not focus on the pH dependence of gas-particle partitioning processes in our revised manuscript.

**Reference**

SPARC performs automated reasoning in chemistry., http://ibmlc2.chem.uga.edu/sparc/index.cfm., 2003.

EPA: Estimation Programs Interface (EPI) Suite ™ for Microsoft®Windows, v4.1, Environmental Protection Agency (EPA), Washington, DC,, 24, 2011.

Ip, H., Huang, X., and Jian, Z. Y.: Effective Henry's law constants of glyoxal, glyoxylic acid, and glycolic acid, Geophysical Research Letters, 36, 2009.

Kampf, C. J., Waxman, E. M., Slowik, J. G., Dommen, J., Pfaffenberger, L., Praplan, A. P., Prevot, A. S., Baltensperger, U., Hoffmann, T., and Volkamer, R.: Effective Henry's law partitioning and the salting constant of glyoxal in aerosols containing sulfate, Environmental Science & Technology, 47, 4236-4244, 10.1021/es400083d, 2013.

Saxena, P. and Hildemann, L. M.: Water-soluble organics in atmospheric particles: A critical review of the literature and application of thermodynamics to identify candidate compounds, Journal of Atmospheric Chemistry, 24, 57-109, 1996.

Shen, H., Chen, Z., Li, H., Qian, X., Qin, X., and Shi, W.: Gas-Particle Partitioning of Carbonyl Compounds in the Ambient Atmosphere, Environmental Science & Technology, 52, 10997-11006, 10.1021/acs.est.8b01882, 2018.

---

## Author Comment (AC3)

**Response to Reviewer #3**

We gratefully thank you for your constructive comments and through review. Below are our point-by-point responses to your comments.

(Q=Question, A=Answer, C=Change in the revised manuscript)

**General Comments:**

Hu et al. present observations of glyoxal and methylglyoxal collected during four seasons in Beijing. The observations included gas-phase and aerosol-phase dicarbonyls. With these observations, the authors investigate the paritioning/reverisible and irreversible uptake of the dicarbonyls. They find that theoretical values underpredict the real-world observations. Further, they find that irreversible uptake dominates in all seasons, though reversible uptake becomes more important in winter time. This study provides an interesting data set and way to investigate this long-standing question of the uptake of dicarbonyls to aerosol as other studies normally just have gas-phase measurements and use a steady state model to derive the first order uptake of glyoxal to aerosol.

Though this paper is of interest to the ACP community, there are some aspects of the paper the authors can improve upon to improve the overall study. With the clarifications suggested below, the manuscript would be acceptable for ACP.

A: We highly appreciate your comments and suggestions. The questions you mentioned are answered as follows.

**Major Comments:**

Q1: One of the major areas that would benefit with expanded text would be the methods. Currently, there is not enough information in order to understand the measurements and discussions from the authors. The following discussions in methods should be added to improve the understanding of the paper:

A1: Thanks for your suggestions and we have expanded the discussion of methods in our revised manuscript. The questions or suggestions you mentioned about the methods are answered as follows.

1a) As the authors are collecting the gas-phase dicarbonyls onto cartridges, a discussion on the percent collected / percent lost both during the collection and extraction / analysis period.

1a): Thanks for your suggestion. Additional field-sampling were launched to estimate the sampling efficiency during the collection of gas-phase dicarbonyls. Two blank DNPH cartridges were connected in tandem to sample the gas-phase dicarbonyls and the sampling conditions were similar to the field observations. Sampling efficiency (SE) were the ratio of dicarbonyl concentrations in the first cartridge to the total concentrations in the two cartridges. And the results were more than 95% for both glyoxal and methylglyoxal, which were equal to the official sampling efficiency provided by Waters Corporation, which is the authoritative corporation to produce DNPH cartridges.

Recovery tests were also conducted using two methods, that was adding standard solutions and repeated extraction. We added additional mixed standard solutions at three spiked levels of 0.025, 0.25 and 2.5μg (namely 50 μL of 0.5 μg·mL$^{-1}$, 5 μg·mL$^{-1}$ and 50 μg·mL$^{-1}$ analytical standards) into the blank DNPH cartridges to determine the carbonyl lost during the extraction and analysis. Then the cartridges were extracted in the same way as the ambient samples. Each group were set with five parallel. The recoveries ranged from 88% to 96% for gas-phase method. Moreover, we also estimated the recovery efficiency by repeated extraction and the recoveries were the ratios of dicarbonyl concentrations in the first extraction to the total concentrations in the two extractions. The results ranged from 92.8% to 99.9%.

1b) Similarly, the authors should have a discussion about the percent collected / percent lost for the dicarbonyl aerosol on filters.

1b): Thanks for your suggestion. Additional field-sampling were also launched to estimate the sampling efficiency during the collection of particle phase of dicarbonyls. We placed a backup Teflon filter after the particle sampling Teflon filter using an independent filter holder to estimate the particle collection efficiency. Both Teflon filters were weighed by a semimicro balance (Sartorius, Germany) to obtain the mass concentration of collected particles. The mass concentrations of particles collected on

the backup filters were closed to zero, indicating that the sampling efficiency of particle were more than 99%.

Similar to gas-phase methods, recovery tests were also conducted using two methods. We adding additional mixed standard solutions at three spiked levels of 0.025, 0.25 and 2.5μg into the blank quartz filters to determine the carbonyl lost during the extraction and analysis. Then the filters were extracted in the same way as the ambient samples. Each group were set with five parallel. The recoveries ranged from 85% to 96% for particle-phase method. Moreover, we also estimated the recovery efficiency by repeated extraction and the recoveries ranged from 90% to 99.9%.

1a), 1b): Lines 147-162 in Sect. 2.3:

Additional field-sampling were launched to estimate the sampling efficiency during the collection. Two blank DNPH cartridges were connected in tandem to assess the sampling efficiency of gas-phase carbonyls. The sampling efficiency of the cartridges were the ratios of dicarbonyl concentrations in the first cartridge to the total concentrations in the two cartridges and the results were more than 95% for both glyoxal and methylglyoxal. Similarly, a backup Teflon filter were placed after the particle sampling Teflon filter using an independent filter holder to estimate the particle collection efficiency. Both Teflon filters were weighed by a semimicro balance (Sartorius, Germany) to obtain the mass concentration of collected particles. The mass concentrations of particles collected on the backup filters were closed to zero, indicating that the sampling efficiency of particle were more than 99%.

Moreover, recovery tests were also conducted using two methods - adding standard solution and repeated extraction. We added the standard solution at three spiked levels of 0.025, 0.25 and 2.5 μg (namely 50 μL of 0.5 μg·mL-1, 5 μg·mL-1 and 50 μg·mL-1 analytical standards) into blank DNPH cartridges and blank quartz filters to determine the carbonyl lost during the extraction and analysis. And then the cartridges and filters were extracted in the same way as the ambient samples. Each group were set with five parallel. The recoveries were ranged from 88% to 96% for gas-phase method and ranged from 85% to 96% for particle-phase method. Moreover, we also estimated the recovery efficiencies by repeated extraction and the recoveries were the ratios of

dicarbonyl concentrations in the first extraction to the total concentrations in the two extractions. The results ranged from 92.8% to 99.9% for gas-phase method and ranged from 90% to 99.9% for particle-phase method.

1c) Another reviewer commented, and I agree, a discussion about potential artifacts for both methods, but especially the aerosol filter collection, needs to be included. This includes if there was a cyclone for size selection, is there a denuder to prevent gas-phase from being collected onto the filters, how long the filters were collected, potential lost of dicarbonyls from the filters during sampling or preparation, and potential side reactions on the filters that may have led to biases.

1c): Thanks for your suggestions. We have expanded the discussion about potential artifacts for both methods. Following measurements were conducted during the sample collection, pretreatment and analysis to ensure the accuracy of results: (1) Before sampling, flow calibration and airtightness tests of sampling devices were conducted, and flow difference were less than 10%; (2) After sampling, the gas-phase samples were resealed by its end cap and plug, and stored in the provided pouch under cool environment (<4°C), the particle-phase samples were stored in the sealed boxes wrapped by pre-baked aluminum foils under freezing environment (<-18°C), both gas-phase and particle-phase samples were extracted and analyzed within a week; (3) The extraction processes were conducted in fume hood with glassware, which was rinsed with acetonitrile for at least three times; (4) A calibration run was performed each day to determine the response factor of the detector and recalibration was performed if the relative deviation of the RF was beyond 5%.

As for aerosol filter collection, we used a four-channel ambient particles sampler (TH-16A, Wuhan Tianhong) with $PM_{2.5}$ cutters ($D_{a50}$=2.5 ± 0.2 μm, $\sigma_g$=1.2 ± 0.1) for size selection. The filters were pre-baked at 550 °C in muffle furnace (Meicheng Corporation) for 6 h to remove impurities before sampling and were continuously collected every 11.5-12 h daily. During the sampling of particle-phase dicarbonyls, we didn't use denuders and the lack of denuders may lead to positive artifacts by adsorption of gas-phase dicarbonyls on the quartz filters. To evaluate and eliminate the adsorption artifacts, we followed the same method described in the artifact experiments by (Shen

et al., 2018), which placed a backup quartz filter after the sampling filter using an independent filter holder. During the sampling, the sampling filters would collect the particles and adsorbed gaseous dicarbonyls, while the backup filters would only collect gaseous dicarbonyls. And the ratio of measured dicarbonyls in second filters to that in the first were lower than 20%, which was equal to the previous study (Shen et al., 2018). Particulate concentration of dicarbonyls used in this study were all corrected by the adsorption artifacts.

As for gas-phase dicarbonyls, besides environmental contamination, ozone in urban air could degrade the hydrazone derivatives and we placed an ozone scrubber cartridge (Waters Corporation) on the inlet of the DNPH cartridge. Moreover, the elution flow could affect the extraction efficiency, which should be less than 3 mL/min.

We have fully discussed the potential artifacts for both methods in Section 2 Materials and Method in our revised manuscript.

1d) Besides how much material is recovered for sampling, how well were these two dicarbonyls identified? E.g., as it is expected that there are other dicarbonyls, how well were the peaks separated for glyoxal and methylglyoxal (an example chromatogram in the SI would be beneificial)?

1d): Thanks for your suggestion. We presented an example chromatogram in SI and the retention time for glyoxal and methylglyoxal were 29 min and 41 min, respectively. The peaks for glyoxal and methylglyoxal were separated effectively with each other. And the peaks for glyoxal and methylglyoxal in extraction of sampling filters were corresponding to those in standard solution.

1d):

[Figure]

Figure S2: The chromatogram for the measured carbonyls. (a) chromatogram for standard solutions of ten measured carbonyls; (b) specific chromatogram for standard solutions of glyoxal and methylglyoxal; (c) chromatogram for extraction of sampling filters (FA: formaldehyde; AA: acetaldehyde; AC: acetone; PA: propionaldehyde; MACR: methacrolein; BA: butyraldehyde; MVK: methyl vinyl ketone; BZA: benzaldehyde; GL: glyoxal; MG: methylglyoxal).

1e) The authors state the assumption that all dicarbonyls that have done reversible partitioning to the aerosol-phase are extracted as the parent compound. A discussion showing this to be true either in the methods or in the results would be beneficial (e.g., if possible, having the reversible products on a filter, extract, and see if they come out as glyoxal/methylglyoxal in the chromatagram).

1e): Thanks for your suggestion. By adding excess derivatization agent (like 2,4-dinitrophenylhydrazone in this study), dicarbonyls as well as their reversibly formed hydrates and oligomers are efficiently transformed into dicarbonyl-bis-2,4-dinitrophenylhydrazone, which was quantified by means of analysis techniques (Kampf et al., 2013). Moreover, Healy et al.(2008) have confirmed that derivatization agent was found to efficiently dissolve a trimeric glyoxal standard and convert the resulting monomers to oxime derivatives, and oligomers were not detected in the extracts of filter samples by GC-MS analysis, indicating the use of excess derivatization agent could convert the hydrates and oligomers back to the monomeric species by removing

dicarbonyl monomers from the extract as soon as they are formed. We have discussed it in Section 3.2 (3.2 Reversible pathways) in our revised manuscript. And we prepare to have the reversible products on a filter or extracts to provide a further proof in our future study.

1e): Lines 271-277 in Sect. 3.2:

By adding excess derivatization agent (like 2,4-dinitrophenylhydrazone in this study), dicarbonyls as well as their reversibly formed products are efficiently transformed into dicarbonyl-bis-2,4-dinitrophenylhydrazone, which are quantified as monomers by means of analysis techniques (Kampf et al., 2013). Moreover, Healy et al.(2008) have confirmed that derivatization agent was found to efficiently dissolve a trimeric glyoxal standard and convert the resulting monomers to oxime derivatives, and oligomers could not be detected in the extracts of filter samples by GC-MS analysis, also indicating the use of excess derivatization agent could efficiently convert the hydrates and oligomers back to the monomeric species by removing dicarbonyl monomers from the extract as soon as they are formed.

1f) The irreversible uptake calculation (page 9, line 252 - page 10, line 268) should be moved to the methods.

1f): We have moved the irreversible uptake calculation to the methods.

1g) Were blanks collected? What is the LOD for both methods?

1g): We have collected the blanks in both gas-phase samples and particle-phase samples. The blank gas-phase samples (blank DNPH cartridge samples) were collected every 3 days by placing it near the gas inlet for the same duration without artificial pumping. And the blank particle-phase samples (blank quartz filters) also were collected every 3 days by placing it on the $PM_{2.5}$ inlet with flow rate of 0 L/min. The data used in this study were all calibrated by blanks.

The LOD are approximately 1 ng/m$^3$ for particle-phase samples and ~50 pptv for gas-phase samples.

1g): Lines 139-146 in Sect. 2.3:

Blank samples were collected every three days and then were stored and extracted by the same procedure as that for ambient samples. The blank gas-phase samples were

collected by placing blank DNPH cartridges near the gas inlet for the same duration without artificial pumping. And the blank particle-phase samples were collected by placing blank quartz filters on the $PM_{2.5}$ inlet with flow rate of 0 L/min. All data used in this study were all calibrated by blanks.

The limit of detection (LOD) of two methods was 50 pptv for gaseous carbonyls and 1 ng·m$^{-3}$ for particulate carbonyls, which is similar to our previous literature (Shen et al., 2018). Sample amount to limit of detection ratios were significantly higher than 1.0 for both gas- and particle-phase samples, indicating that the sensitivity of the methods was sufficient to analyze the samples.

1h) What is the uncertainty associated with the assumptions made to calculate Kp? E.g., there would be high uncertainty in activity coefficient, vapor pressure, and potentially the absorbing fraction of the total particulate matter, depending on how well the methods measured total OA.

1h): Thanks for your suggestion and we have added the discussion of uncertainties in $K_p^t$ calculations in our revised Supporting Information as follows:

1h): Lines 13-31 in Supporting Information:

According to Eq.2, there would be uncertainty in temperature (T), activity coefficient ($\zeta$), vapor pressure($p_L^0$), absorbing fraction of the total particulate matter (fom), and molecular weight of the organic phase (MWOM) to calculate the theoretical partitioning coefficients ( $K_p^t$ ). The $K_p^t$ values could increase with increasing temperature T, which ranged from 265.53 K to 310.75 K in our observations. We calculate the $K_p^t$ at the two extreme temperature and the ratios $K_{p,\ 310.75K}^t/K_{p,\ 265.53K}^t$ are always lower than 3 for both glyoxal and methylglyoxal. Aerosol phase activity coefficients $\zeta$ are a function of the aerosol composition (Jang et al., 1997; Seinfeld et al., 2001) but are thought to modify $K_p^t$ by less than one order of magnitude (Bowman and Melton, 2004). And fom was little changed during our observations and was usually within ~0.3-0.6 in urban Beijing (Huang et al., 2014; Ma et al., 2022). As for MW$_{OM}$, previous laboratory experiments have shown that the molecular weight of individual

constituents in isoprene SOA and 1,3,5-TMB SOA ranges from 100 to 1000 and that the average molecular weight increases with aerosol age due to oligomerization (Kalberer and M., 2004; Dommen et al., 2006). We used a MWOM value of 200 g·mol-1, which is used in previous work (Barsanti and Pankow, 2004; Williams et al., 2010; Shen et al., 2018) and increasing MWOM even to 500 g·mol-1 would only reduce $K_p^t$ by a factor of 4. Moreover, we calculated the vapor pressure by the extended aerosol inorganic model (E-AIM, http://www.aim.env.uea.ac.uk/aim/ddbst/pcalc_main.php) (Clegg et al., 1998), the results of which were close to those of other methods, such as the SPARC online calculator (Version 3.1) (Hilal et al., 1995; Healy et al., 2008). Overall, the uncertainties associated with $K_p^t$ calculation were within one order of magnitude. However, as discussed in our study, the discrepancy between field-measured partitioning coefficients and the theoretical ones was more than five orders of magnitude and the uncertainty in the $K_p^t$ calculation would not change the interpretations and conclusions of this work.

Q2: As the authors state different comparisons for the values they observed/calculated, it would be beneficial to either in their current tables or in a new table compare their results with literature.

A2: Thanks for your suggestion. We compared the $K_p^f/K_p^t$ values (field-measured partitioning coefficients/theoretical partitioning coefficients) in this study and the values published in previous literatures in a new table in Supplement Information as follows.

C2:

**Table S2:** Comparison between $K_p^f/K_p^t$ values (field-measured partitioning coefficients/theoretical partitioning coefficients) in this study and the values published in previous literatures.

| Dicarbonyls | Seasons | $K_p^f$ | $K_p^f/K_p^t$ | Reference |
|---|---|---|---|---|
| Glyoxal | summer | $3.23 \times 10^{-4}$ | $8.90 \times 10^5$ | Qian et al.(2019) |
| | | $6.31 \times 10^{-4}$ | $1.32 \times 10^6$ | Cui et al.(2021) |
| | | $1.02 \times 10^{-1}$ | / | Ortiz et al.(2013) |
| | | $8.11 \times 10^{-4}$ | $2.48 \times 10^6$ | This study |
| | winter | $1.44 \times 10^{-3}$ | $1.13 \times 10^6$ | Shen et al.(2018) |
| | | $1.71 \times 10^{-3}$ | $1.05 \times 10^6$ | Cui et al.(2021) |
| | | $1.30 \times 10^{-2}$ | $1.02 \times 10^7$ | This study |
| | spring | $1.33 \times 10^{-3}$ | $1.80 \times 10^6$ | Cui et al.(2021) |
| | | $1.43 \times 10^{-2}$ | $3.55 \times 10^7$ | This study |
| | autumn | $1.05 \times 10^{-3}$ | $1.26 \times 10^6$ | Cui et al.(2021) |
| | | $2.14 \times 10^{-3}$ | $3.41 \times 10^6$ | This study |
| Methylglyoxal | summer | $4.07 \times 10^{-5}$ | $5.22 \times 10^4$ | Qian et al.(2019) |
| | | $1.40 \times 10^{-4}$ | $1.31 \times 10^5$ | Cui et al.(2021) |
| | | $7.41 \times 10^{-2}$ | / | Ortiz et al.(2013) |
| | | $1.49 \times 10^{-4}$ | $2.10 \times 10^5$ | This study |
| | winter | $4.19 \times 10^{-4}$ | $1.53 \times 10^5$ | Shen et al.(2018) |
| | | $4.27 \times 10^{-4}$ | $1.16 \times 10^5$ | Cui et al.(2021) |
| | | $2.60 \times 10^{-3}$ | $9.93 \times 10^5$ | This study |
| | spring | $3.48 \times 10^{-4}$ | $2.10 \times 10^5$ | Cui et al.(2021) |
| | | $1.06 \times 10^{-3}$ | $8.77 \times 10^5$ | This study |
| | autumn | $2.07 \times 10^{-4}$ | $1.11 \times 10^5$ | Cui et al.(2021) |
| | | $9.55 \times 10^{-4}$ | $7.07 \times 10^5$ | This study |

Q3: I agree with the other reviewers that the discussion of theory (Section 3.1.2) does not add much to the paper as this is generally already known and would advise to either reduce this discussion or potentially remove it for more room to expand upon the reversible, irreversible, and methods.

A3: Thanks for your suggestion. We have removed the discussion of discrepancy between the field measured coefficients and the theoretical ones to Supplement Information.

Q4: It is currently unclear how the authors are separating irreversible and reversible. This is especially important in the partitioning calculations, as how much could the irreversible uptake be influencing the calculated value? Further, as the reversible was 10% or less the process the dicarbonyls undergone, is that within the associated uncertainty in the calculations, indicating potentially minimal reversible lost?

A4: Gas-particle process of dicarbonyls in this study is separated into reversible and irreversible pathways, and the separation is based on the reversibility of chemical reaction of dicarbonyls occurred on condensed phase (Ervens and Volkamer, 2010; Kampf et al., 2013; Ling et al., 2020; Galloway et al., 2008). The reversible pathways here include hydration, dimerization and oligomerization, the product of which could revert to their parent compounds during extraction and could be directly detected in particle-phase samples. And the irreversible pathways here refer to irreversible uptake, which is driven by oxidative compounds in aerosols. As glyoxal/methylglyoxal + OH chemistry plays a dominant role in irreversible pathways, we use uptake coefficients $\gamma$ (Eq. 4-7) to estimate and quantify the processes. Although the products of irreversible pathways could not be directly detected in particle phase and don't contribute to the $c_p$ term in partitioning coefficient calculation (Eq.1), the irreversible pathways can well explain the overestimation of modeled dicarbonyl mixing ratios, which is about 3-6 times higher than the observed ones (Volkamer et al., 2007; Ling et al., 2020). Moreover, the reversible pathways were 10% or less the process the dicarbonyls undergone and the associated uncertainties in the calculations were ~5%, indicating the minimal reversible lost was closed to 1-2%.

C4: Lines 85-86 in Sect.1:

These processes are divided into reversible pathways and irreversible pathways, which is based on the reversibility of chemical reaction of dicarbonyls occurred on condensed phase (Ervens and Volkamer, 2010; Kampf et al., 2013; Ling et al., 2020; Galloway et al., 2008).

Although the products of irreversible pathways could not be directly detected in particle phase and didn't directly contribute to the increase of particulate dicarbonyls, the irreversible pathways could contribute to the decrease of gasous dicarbonyls and well explain the overestimation of modeled dicarbonyl mixing ratios, which was about 3-6 times higher than the observed ones (Volkamer et al., 2007; Ling et al., 2020).

Q5: It is currently unclear how the authors derived the values in Fig. 2 (reversible pathway with units of ng/ug). If this is from one of the equations, please state and that will help better understand where the data from this figure originated from. If something else, please describe.

A5: Thanks for your suggestion. Gas-phase dicarbonyls could partition into aerosol liquid water by dissolution, and then reversibly form hydrates and oligomers. Both dissolves dicarbonyl monomers and reversibly formed production are efficiently transformed into glyoxal-bis-2,4-dinitrophenylhydrazone, which was quantified by means of HPLC-UV in this study. The concentrations of dissolved dicarbonyl monomers were estimated using Henry's law coefficients, which is used to determine the physical solubility of carbonyls (e.g., $K_H$=5 M·atm$^{-1}$ for glyoxal) (Schweitzer et al., 1998). The results were negligible compared to the concentrations of carbonyls in hydrate and oligomer forms. Therefore, the data in Figure 2, which referred to particle-phase concentration of dicarbonyls in reversible partitioning pathways, were equal to the measured concentration of carbonyls by HPLC-UV. We have described it in our manuscript.

C5: Lines 277-283 in Sect. 3.2:

Both dissolves dicarbonyl monomers and reversibly formed production are efficiently transformed into carbonyl-bis-2,4-dinitrophenylhydrazone, which was quantified by means of HPLC-UV in this study. The concentrations of dissolved dicarbonyl monomers were estimated using Henry's law coefficients, which is used to determine

the physical solubility of carbonyls (e.g., $K_H$=5 M·atm$^{-1}$ for glyoxal) (Schweitzer et al., 1998). The results were negligible compared to the concentrations of carbonyls in hydrate and oligomer forms. Thus, the concentrations of particle-phase dicarbonyl in reversible partitioning pathways were close to the measured concentration of carbonyls by HPLC-UV.

Q6: Another concern with Fig. 2 is the fact the authors are showing trends vs pH. As they are calculating their pH from only aerosol-phase measurements, there is large inherant uncertainty in the pH values as there is no gas-phase measurements to constrain the partitioning of the semi-volatile gases (NH$_3$, HNO$_3$, or HCl), which can lead to large deviations in the calculated pH from real world observations. I strongly advised the authors to not use the pH as it does not add much to the results.

A6: Thanks for your suggestion. The pH values used in this study indeed contains large inherent uncertainty and we have deleted the discussion of pH in our revised manuscript.

Q7: I'm assuming the values listed in Table S2 are for bulk-phase reactions instead of aerosol-phase reactions. Recent studies have shown that these bulk-phase reactions may not represent the aerosol-phase reaction rates due to the differences in the ionic strength. Therefore, for lines 210 - 222 and Fig. 2b, I would recommend the authors to be careful with those numbers in being the "definitive" product (also correct porduct to product in 2b) distribution to the potential product distribution with uncertainty due to bulk vs aerosol phase.

A7: Thanks for your suggestion and we regret for the incomprehensive consideration. Compared to bulk phase, the higher ionic strength in aerosol phase could affect aerosol-phase reactions in the following ways: (1) change aerosol hygroscopicity, surface tension and viscosity (Kampf et al., 2013; Sareen et al., 2010); (2) the presence of inorganic ions and corresponding change in the activity of water could shift the hydration equilibrium of organics with multiple hydration states (Loeffler et al., 2006); (3) influence the partitioning process of organics to the condensed phase via salting effects, as the presence of inorganic ions could catalyze and participate in oligomerization reactions (Sareen et al., 2010; McNeill, 2015). However, the reversible aerosol-phase reactions of volatile organic compounds in aerosols have not been

systematically explored due to the lack of suitable reaction parameters and mechanisms. Recently, many laboratory experiments and model simulations have been conducted to evaluate the oligomerization in different seed aerosols. Take glyoxal for example, the dimensionless equilibrium constant of oligomerization $K_{olig}$ varied a lot in different seed aerosols with the values of 1000 on $(NH_4)_2SO_4$ seed particles (Ervens and Volkamer, 2010), >700 on $Na_2SO_4$ seed particles (Corrigan et al., 2008) and 3-5 on NaCl seed particles (Ip et al., 2009). All of these parameters were worth to be considered in reversible aerosol-phase reactions simulation. Moreover, Elrod et al.(2021) investigated the carbonyl hydration equilibria on aqueous atmospheric particles using nuclear magnetic resonance (NMR) spectroscopy coupled with the MG2MS electronic structure method and revealed that the hydration equilibrium constants were ~1000 for glyoxal and ~100 for methylglyoxal. However, the rate constants of glyoxal and methylglyoxal in aerosol-phase have not been accurately quantified and we have to roughly estimate the product distribution on the basis of mechanisms in bulk-phase. The RH dependence of product distribution and the order of magnitude of estimated $K_p$ values were close to those in aerosol-phase. We have added the uncertainty description in product simulation and deleted the definitive numbers in our revised manuscript.

C7: Lines 309-313 in Sect. 3.2:

However, the product distribution here was simulated based on the bulk-phase mechanisms and higher ionic strength in aerosol phase would influence reaction equilibria and rate constant (Ervens and Volkamer, 2010; Mcneill, 2015). The lack of quantitative reaction rate could contribute more uncertainties to the simulation, whereas, the RH dependence of product distribution and the order of magnitude of estimated $K_p$ values were close to those in aerosol-phase and the roughly simulation could help to understand the reversible partitioning pathways of dicarbonyls.

Q8: Lines 276-279: It's currently unclear how the authors are drawing the conclusion that methylglyoxal is exhibiting unexpected salting-in effects if they are using Eq. 4 - 7 to calculate the uptake coefficient. As these equations don't include the aerosol

composition or ionic strength, further clarification on this conclusion would help this statement.

A8: Thanks for your suggestion and we regret for the unclear expression. Methylglyoxal always presents a "salting-out" effect because of its increasing steric hindrance in ion hydration shell in previous laboratory studies (Waxman et al., 2015). However, in this study, methylglyoxal exhibited an unexpected "salting-in" effect in ambient particles due to much more complex compositions and higher ionic strength in ambient particles, which was also reported in other observational studies (Shen et al., 2018; Cui et al., 2021). Figure S4 presents the Setschenow plot of dicarbonyls versus aqueous sulfate, nitrate, and ammonia (SNA) concentration in aerosol. The negative salting constant indicated the "salting in" effects, which could result in exponential solubility and higher Henry's law coefficient values for methylglyoxal in the real atmospheric. Although the equations (Eq.4-7) calculating uptake coefficients don't include the aerosol composition or ionic strength, the higher effective Henry's law coefficient values (eff $K_H$) in Eq.4 could lead to higher uptake coefficient values in this study comparing to other experimental ones (Curry et al., 2018; De Haan et al., 2018). Moreover, Li et al.(2021) pointed out that methylglyoxal is more reactive and have larger uptake coefficient on seed particles under atmospherically relevant concentrations.

C8: Lines 350-357 in Sect. 3.3.1:

Moreover, uptake coefficients for methylglyoxal were with an average value of $2.0 \times 10^{-3}$ and were higher than those reported in other experimental studies, which varied from $10^{-6}$ to $10^{-3}$ (Curry et al., 2018; De Haan et al., 2018). On the one hand, conflicting with previous experimental results (Waxman et al., 2015), methylglyoxal exhibited an unexpected salting-in effect in real atmosphere due to much more complex compositions and higher ionic strength in ambient particles, which was also reported in other observational studies (Shen et al., 2018; Cui et al., 2021). And the higher Henry's law coefficient values in Eq.4 could lead to higher uptake coefficient values. One the other hand, a recent study also provided direct experimental evidence to confirm that

methylglyoxal is more reactive and have larger uptake coefficients on seed particles under atmospherically relevant concentrations (Li et al., 2021).

**Minor and Technical Comments:**

Q1: Since ionic strength is being calculated with the aerosol liquid water, it may be useful to look at how these parameters relate to ionic strength.

A1: Thanks for your suggestion. We have expanded the discussion about the relationship between reversible/irreversible pathways of dicarbonyls and ionic strength in wet aerosol as follows.

C1: Lines 293-297 in Section 3.2:

Ionic strength could also influence the reversible partitioning process as it is closely related to aerosol liquid water and RH conditions. The presence of inorganic ions could catalyze and participate in oligomerization reactions via salting effects (Sareen et al., 2010; Mcneill, 2015). Whereas, increasing viscosity of particles with increasing ionic strength could slow down all particle-phase reactions, and the reversible nucleophilic addition of inorganic ions (e.g., sulfate ions) at carbonyl carbons deactivates the molecule for further oligomerization (Kampf et al., 2013).

Lines 383-386 in Sect. 3.3.2:

The particulate concentration of dicarbonyls via irreversible pathways generally decreased with increasing RH. Concentrated inorganic solutions and relatively higher ionic strength in aerosol water under low RH conditions could jointly contribute to the hydration of dicarbonyls, the products of which could easily participate into the following radical reactions via H-abstraction.

Q2: For Fig. 1, it is currently hard to following what is happening with the particulate-phase dicarbonyls. I would recommend including a thin-line connecting the points to better see the data and potential trends.

A2: Thanks for your suggestion. We have added a thin-line to connect the points in Figure 1 in our revised manuscript as follows.

C2:

[Figure]

Figure 1e: Time series of meteorological parameters and gas- and particle-phase glyoxal and methylglyoxal observed in spring.

Q3: Line 287, it should be Fig. S8 instead of S7.

A3: Thanks for your suggestion. We have corrected it in our revised manuscript.

Q4: For Fig. S7, it is unclear what sequence number (x-axis) and what the grey shaded area are for.

A4: Thanks for your suggestion and we regret for the unclear expression. The sequence number (x-axis) refers to the serial number of samples. And the grey shaded area refers to the variation range of modeled oxalate concentrations, which is constrained by OH concentrations in aerosol liquid water. And we have specified it in the caption of Fig. S7 in our revised manuscript.

Q5: For table 1, it would be useful to include the dates of the measurements.

A5: Thanks for your suggestion and we have added the dates of the measurements in Table 1 in our revised manuscript.

Q6: For table 2, it is unclear how "theory" Henry's law constant is calculated compared to the "field" values.

A6: We regret for the unclear expression. The theoretical Henry's law constant refers to the Henry's law constant of dicarbonyls in pure water, which is calculated from Eq. 1-2 in previous studies (Sander, 2015). We have specified it in our revised manuscript.

$$\frac{d(\ln H)}{d(1/T)} = -\frac{-\Delta_{sol}H}{R} \qquad (1)$$

$$H(T) = H^{\ominus} \times \exp\left(\frac{-\Delta_{sol}H}{R} \times \left(\frac{1}{T} - \frac{1}{T^{\ominus}}\right)\right) \qquad (2)$$

Where H(T) (mol·m$^{-3}$·Pa$^{-1}$) is the Henry's law constant of dicarbonyls in pure water at different ambient temperature T(K); H$^{\Theta}$(mol·m$^{-3}$·Pa$^{-1}$) is the Henry's law constant of dicarbonyls in pure water at standard temperature T$^{\Theta}$(298.15K); $\Delta_{sol}H$ (J/mol) is molar enthalpy of dissolution. For glyoxal, H$^{\Theta}$ is 4100 mol·m$^{-3}$·Pa$^{-1}$, $\frac{-\Delta_{sol}H}{R}$ is 7500 K (Ip et al., 2009); and for methylglyoxal, H$^{\Theta}$ is 34 mol·m$^{-3}$·Pa$^{-1}$, $\frac{-\Delta_{sol}H}{R}$ is 7500 K (Betterton and Hoffmann, 1988).

**Reference:**

Barsanti, K. C. and Pankow, J. F.: Thermodynamics of the formation of atmospheric organic particulate matter by accretion reactions—Part 1: aldehydes and ketones, Atmospheric Environment, 38, 4371-4382, 2004.

Betterton, E. A. and Hoffmann, M. R.: Henry's law constants of some environmentally important aldehydes, Environmental Science & Technology, 22, 1415-1418, 1988.

Bowman, F. M. and Melton, J. A.: Effect of activity coefficient models on predictions of secondary organic aerosol partitioning, Journal of Aerosol Science, 35, 1415-1438, 10.1016/s0021-8502(04)00286-1, 2004.

Clegg, S. L., Brimblecombe, P., and Wexler, A. S.: Thermodynamic Model of the System H+-NH4+-SO42--NO3--H2O at Tropospheric Temperatures, Journal of Physical Chemistry A, 102, 2137-2154, 1998.

Corrigan, A. L., Hanley, S. W., and Haan, D. D.: Uptake of glyoxal by organic and Inorganic aerosol, Environmental Science & Technology, 42, 4428, 2008.

Cui, J., Sun, M., Wang, L., Guo, J., Xie, G., Zhang, J., and Zhang, R.: Gas-particle partitioning of carbonyls and its influencing factors in the urban atmosphere of Zhengzhou, China, Science of the Total Environment, 751, 142027, 10.1016/j.scitotenv.2020.142027, 2021.

Curry, L. A., Tsui, W. G., and McNeill, V. F.: Technical note: Updated parameterization of the reactive uptake of glyoxal and methylglyoxal by atmospheric aerosols and cloud droplets, Atmospheric Chemistry and Physics, 18, 9823-9830, 10.5194/acp-18-9823-2018, 2018.

De Haan, D. O., Jimenez, N. G., de Loera, A., Cazaunau, M., Gratien, A., Pangui, E., and Doussin, J. F.: Methylglyoxal Uptake Coefficients on Aqueous Aerosol Surfaces, Journal of Physical Chemistry A, 122, 4854-4860, 10.1021/acs.jpca.8b00533, 2018.

Dommen, J., Metzger, A., Duplissy, J., Kalberer, M., and Baltensperger, U.: Laboratory observation of oligomers in the aerosol from isoprene/NOx photooxidation, Geophysical Research Letters, 33, 2006.

Elrod, M. J., Sedlak, J. A., and Ren, H.: Accurate Computational Model for the Hydration Extent of Atmospherically Relevant Carbonyls on Aqueous Atmospheric Particles, ACS Earth and Space Chemistry, 5, 348-355,

10.1021/acsearthspacechem.0c00322, 2021.

Ervens, B. and Volkamer, R.: Glyoxal processing by aerosol multiphase chemistry: towards a kinetic modeling framework of secondary organic aerosol formation in aqueous particles, Atmospheric Chemistry and Physics, 10, 8219-8244, 10.5194/acp-10-8219-2010, 2010.

Galloway, M. M., Chhabra, P. S., Chan, A., Surratt, J. D., and Keutsch, F. N.: Glyoxal uptake on ammonium sulphate seed aerosol: reaction products and reversibility of uptake under dark and irradiated conditions, Atmospheric Chemistry and Physics, 8, 2008.

Healy, R. M., Wenger, J. C., Metzger, A., Duplissy, J., Kalberer, M., and Dommen, J.: Gas/particle partitioning of carbonyls in the photooxidation of isoprene and 1,3,5-trimethylbenzene, Atmospheric Chemistry and Physics, 8, 2008.

Hilal, S. H., Karickhoff, S. W., and Carreira, L. A.: A Rigorous Test for SPARC's Chemical Reactivity Models: Estimation of More Than 4300 Ionization pKas, Quantitative Structure-Activity Relationships, 14, 348-355, 1995.

Huang, R. J., Zhang, Y., Bozzetti, C., Ho, K. F., Cao, J. J., Han, Y., Daellenbach, K. R., Slowik, J. G., Platt, S. M., Canonaco, F., Zotter, P., Wolf, R., Pieber, S. M., Bruns, E. A., Crippa, M., Ciarelli, G., Piazzalunga, A., Schwikowski, M., Abbaszade, G., Schnelle-Kreis, J., Zimmermann, R., An, Z., Szidat, S., Baltensperger, U., El Haddad, I., and Prevot, A. S.: High secondary aerosol contribution to particulate pollution during haze events in China, Nature, 514, 218-222, 10.1038/nature13774, 2014.

Ip, H., Huang, X., and Jian, Z. Y.: Effective Henry's law constants of glyoxal, glyoxylic acid, and glycolic acid, Geophysical Research Letters, 36, 2009.

Jang, M., Kamens, R. M., Leach, K. B., and Strommen, M. R.: A Thermodynamic Approach Using Group Contribution Methods to Model the Partitioning of Semivolatile Organic Compounds on Atmospheric Particulate Matter, Environmental Science & Technology, 31, 2805-2811, 1997.

Kalberer and M.: Identification of polymers as major components of atmospheric organic aerosols, Science, 303, 1659-1662, 2004.

Kampf, C. J., Waxman, E. M., Slowik, J. G., Dommen, J., Pfaffenberger, L., Praplan, A. P., Prevot, A. S., Baltensperger, U., Hoffmann, T., and Volkamer, R.: Effective Henry's law partitioning and the salting constant of glyoxal in aerosols containing sulfate, Environmental Science & Technology, 47, 4236-4244, 10.1021/es400083d, 2013.

Li, Y., Ji, Y., Zhao, J., Wang, Y., Shi, Q., Peng, J., Wang, Y., Wang, C., Zhang, F., Wang, Y., Seinfeld, J. H., and Zhang, R.: Unexpected Oligomerization of Small alpha-Dicarbonyls for Secondary Organic Aerosol and Brown Carbon Formation, Environmental Science & Technology, 55, 4430-4439, 10.1021/acs.est.0c08066, 2021.

Ling, Z., Xie, Q., Shao, M., Wang, Z., Wang, T., Guo, H., and Wang, X.: Formation and sink of glyoxal and methylglyoxal in a polluted subtropical environment: observation-based photochemical analysis and impact evaluation, Atmospheric Chemistry and Physics, 20, 11451-11467, 10.5194/acp-20-11451-2020, 2020.

Loeffler, K. W., Koehler, C. A., Paul, N. M., and Haan, D. D.: Oligomer formation in evaporating aqueous glyoxal and methyl glyoxal solutions, Environmental Science & Technology, 40, 6318, 2006.

Ma, J., Ungeheuer, F., Zheng, F., Du, W., Wang, Y., Cai, J., Zhou, Y., Yan, C., Liu, Y., Kulmala, M., Daellenbach, K. R., and Vogel, A. L.: Nontarget Screening Exhibits a Seasonal Cycle of PM2.5 Organic Aerosol Composition in Beijing, Environ Sci Technol, 10.1021/acs.est.1c06905, 2022.

McNeill, V. F.: Aqueous organic chemistry in the atmosphere: sources and chemical processing of organic aerosols, Environ Sci Technol, 49, 1237-1244, 10.1021/es5043707, 2015.

Ortiz, R., Shimada, S., Sekiguchi, K., Wang, Q., and Sakamoto, K.: Measurements of changes in the atmospheric partitioning of bifunctional carbonyls near a road in a suburban area, Atmospheric Environment, 81, 554-560, 10.1016/j.atmosenv.2013.09.045, 2013.

Qian, X., Shen, H., and Chen, Z.: Characterizing summer and winter carbonyl compounds in Beijing atmosphere, Atmospheric Environment, 214, 10.1016/j.atmosenv.2019.116845, 2019.

Sander, R.: Compilation of Henry's law constants (version 4.0) for water as solvent, Atmospheric Chemistry and Physics, 15, 4399-4981, 10.5194/acp-15-4399-2015, 2015.

Sareen, N., Schwier, A. N., Shapiro, E. L., Mitroo, D., and Mcneill, V. F.: Secondary organic material formed by methylglyoxal in aqueous aerosol mimics, Atmospheric Chemistry and Physics, 10, 997-1016, 2010.

Schweitzer, F., Magi, L., Mirabel, P., and George, C.: Uptake Rate Measurements of Methanesulfonic Acid and Glyoxal by Aqueous Droplets, Journal of Physical Chemistry A, 102, 593-600, 1998.

Seinfeld, J. H., Erdakos, G. B., Asher, W. E., and Pankow, J. F.: Modeling the Formation of Secondary Organic Aerosol (SOA). 2. The Predicted Effects of Relative Humidity on Aerosol Formation in the α-Pinene-, β-Pinene-, Sabinene-, Δ3-Carene-, and Cyclohexene-Ozone Systems, Environmental Science & Technology, 35, 1806-1817, 2001.

Shen, H., Chen, Z., Li, H., Qian, X., Qin, X., and Shi, W.: Gas-Particle Partitioning of Carbonyl Compounds in the Ambient Atmosphere, Environmental Science & Technology, 52, 10997-11006, 10.1021/acs.est.8b01882, 2018.

Volkamer, R., San Martini, F., Molina, L. T., Salcedo, D., Jimenez, J. L., and Molina, M. J.: A missing sink for gas-phase glyoxal in Mexico City: Formation of secondary organic aerosol, Geophysical Research Letters, 34, 10.1029/2007gl030752, 2007.

Waxman, E. M., Elm, J., Kurten, T., Mikkelsen, K. V., Ziemann, P. J., and Volkamer, R.: Glyoxal and Methylglyoxal Setschenow Salting Constants in Sulfate, Nitrate, and Chloride Solutions: Measurements and Gibbs Energies, Environmental Science & Technology, 49, 11500-11508, 10.1021/acs.est.5b02782, 2015.

Williams, B. J., Goldstein, A. H., Kreisberg, N. M., and Hering, S. V.: In situ measurements of gas/particle-phase transitions for atmospheric semivolatile

organic compounds, Proceedings of the National Academy of Sciences, 107, 6676-6681, 10.1073/pnas.0911858107, 2010.

---

## Author Comment (AC4)

**Response to Reviewer #4**

We gratefully thank you for your constructive comments and thorough review. Below are our point-by-point responses to your comments.

(Q=Question, A=Answer, C=Change in the revised manuscript)

**General Comments:**

The article by Hu et al. titled "Reversible and irreversible gas-particle partitioning of dicarbonyl compounds observed in the real atmosphere" discusses the importance of reversable and irreversible gas-to-particle partitioning of glyoxal and methyl glyoxal. The authors present experimental and modeling results showing how irreversible gas-to-particle partitioning dominates the two partitioning pathways and also highlighted the other reaction processes that were not taken into account in the analysis of this study. The study is relevant for the atmospheric community and can be accepted to ACP after the comments have been addressed.

A: We highly appreciate your comments and suggestions. The questions you mentioned are answered as follows.

**Major Comments:**

Q1: Page 5, line 134: What are the other carbonyls that were measured in the gas and particle phases?

A1: We measured ten carbonyls in gas phase, including formaldehyde, acetaldehyde, acetone, propionaldehyde, methacrolein, butyraldehyde, methyl vinyl ketone, benzaldehyde, glyoxal, and methylglyoxal. And we also measure six carbonyls in particle phase, including formaldehyde, acetaldehyde, acetone, propionaldehyde, glyoxal, and methylglyoxal. We have specified it in our revised manuscript.

C1: Lines 208-211 in Sect. 3.1.1:

Ten carbonyls were measured in the gas phase, including formaldehyde, acetaldehyde, acetone, propionaldehyde, methacrolein, butyraldehyde, methyl vinyl ketone, benzaldehyde, glyoxal, and methylglyoxal; and six carbonyls were measured in the particle phase, including formaldehyde, acetaldehyde, acetone, propionaldehyde, glyoxal, and methylglyoxal.

Q2: Based on what's written at the end of page 7 and later, the measured dicarbonyls in the particle phase are only ones that have formed products of the reversible pathways. It wasn't clear earlier when you talk about experimental partitioning coefficients that they only include reversible partitioning, point it our somewhere earlier to avoid confusion.

A2: Thanks for your suggestion. Previous studies have pointed out that both glyoxal and methylglyoxal are always in hydrate forms or oligomer forms under atmospheric conditions (Barsanti and Pankow, 2005; Liggio et al., 2005; Elrod et al., 2021; Michailoudi et al., 2021). And dicarbonyls in monomer forms only accounts for ~1% in our study with lower physical solubility of dicarbonyls (e.g.,  $K_H=5 \text{ M} \cdot \text{atm}^{-1}$  for glyoxal) (Schweitzer et al., 1998). Most of dissolved dicarbonyl monomers could participate into chemical reactions, forming hydrates and oligomers. We have pointed it in our revised manuscript to avoid confusion as follows.

C2: Lines 170-172 in Sect. 2.3

 $C_p (\mu g \cdot m^{-3})$  is the concentrations of dicarbonyls in the particle phase which is derived from the analysis of extracts, including monomers and their reversibly formed products (the product distribution is discussed in Section 3.2).

Q3: Page 8, line 210: How exactly are the proportions of hydrates and oligomers at different RH calculated? Table S2 gives the hydration rate constants as (pseudo) first order rate constants, so the amount of water should have no effect on the equilibrium, right? Or are the experiments used in these calculations somehow? Please specify in the text.

A3: Thanks for your suggestion. The proportions of hydrates and oligomers are calculated on the basis of the kinetic mechanisms listed in Table S2 using a 0-D box model with a steady-state approach. And the amount of water would have no effet on the equilibrium. We have specified the calculation in our revised manuscript.

C3: Lines 298-300 in Sect. 3.2

To roughly estimate the product distribution of the reversible pathway in the real atmosphere, we simplified reaction mechanisms and calculated the product distribution on the basis of on the basis of the kinetic mechanisms listed in Table S3 using a 0-D box model with a steady-state approach.

**Minor and Technical Comments:**

Q1: In the abstract "These two pathways of dicarbonyls jointly contributed to more than 25% of SOAs in the real atmosphere"

A1: Thanks for your suggestion. We have rephrased the sentences as follow.

C1: Lines 21-22 in the Abstract:

The partitioning processes of dicarbonyls in reversible and irreversible pathways jointly contributed to more than 25% of SOA formation in the real atmosphere.

Q2: Page 1, line 27-28: "The  $\alpha$ -dicarbonyl functionality increases their water solubility and reactivity more than expected" would be better if you say something like "The  $\alpha$ dicarbonyl functionality leads to higher water solubility and reactivity than expected." Otherwise, specify how the solubility and reactivity have increased (from what).

A2: Thanks for your suggestion. The  $\alpha$ -dicarbonyl functionality is hydrophilic and contributes to hydrate formation. The EPA's chemical and physical property calculator, EPISUITE, predicts that the hydrated form of carbonyls is less volatile and more water-soluble than the un-hydrated form (EPA., 2012), owing to the strong effect of the two hydrogen-bonding groups in the hydrated form (Elrod et al., 2021). And hydrate form of carbonyls can easily participate in continuous radical reactions with higher activity by H-abstraction to form higher-molecular-weight oligomers (Michailoudi et al., 2021). We have revised the sentences in the revised manuscript.

C2: Lines 28-33 in Sect. 1:

The  $\alpha$ -dicarbonyl functionality leads to higher water solubility and reactivity of dicarbonyls than expected, as the  $\alpha$ -dicarbonyl functionality is hydrophilic and contributes to hydrate formation. The hydrate form of carbonyls is less volatile and more water-soluble than the un-hydrated form (EPA., 2012), owing to the strong effect of the two hydrogen-bonding groups in the hydrated form (Elrod et al., 2021). Moreover, hydrates can easily participate in radical reactions with higher activity by H-abstraction to form higher-molecular-weight oligomers (Michailoudi et al., 2021).

Q3: page 2, line 36-37: "however, there is still a missing sink for the two dicarbonyls" Do you mean that the known sinks listed before are not large enough to explain the loss of the dicarbonyls from the gas phase? Or that there is a specific sink mentioned by Volkamer et al. that wasn't listed here? Please specify.

A3: Thanks for your suggestion and we regret for the unclear expression. We mean that there is a specific sink mentioned by Volkamer et al. (2007) that wasn't listed before. And the missing sink stated here refers to the gas-particle partitioning process of dicarbonyls. We have specified this sentence in the revised manuscript.

C3: Lines 42-43 in Sect.1:

however, there is still a missing sink for the two dicarbonyls (Volkamer et al., 2007), that's the gas-particle partitioning process, which would be fully discussed in this study.

Q4: page 3, line 72: "among key regions with relatively higher PM2.5 concentrations" Do you mean that the key regions have relatively higher PM2.5? Or Beijing has relative higher PM2.5 concentrations than the other key regions? Please specify.

A4: Thanks for your suggestion and we regret for the unclear expression. We mean that the key regions (including Beijing) have relatively higher PM2.5. We have specified this sentence in the revised manuscript.

C4: Lines 82-83 in Sect.1:

Chen et al. (2021) found that the average concentration of dicarbonyls in Beijing is lowest among the key regions that have relatively higher  $PM_{2.5}$  concentrations, indicating there is a more efficient partitioning process of dicarbonyls.

Q5: Page 5, line 145: define GL and MG

A5: Thanks for your suggestion. GL and MG are the abbreviation of glyoxal and methylglyoxal. And we have defined GL and MG in our revised manuscript.

Q6: Page 6, line 164: "lower temperature promoted the partitioning processes" do you mean gas-to-particle partitioning, or also particle-to-gas? It isn't clear by saying "partitioning processes".

A6: Thanks for your suggestion and we regret for the unclear expression. We mean that lower temperature promoted the gas-to-particle partitioning processes. We have specified this sentence in the revised manuscript.

C6: Line 241 in Section 3.21.2:

lower temperature promoted the gas-to-particle partitioning processes.

Q7: Page 7, line 197: "which are more reactive than their counterparts" how do you determine "more reactive"? Aren't glyoxal and methylglyoxal also reactive, because they quickly react with water to become hydrates? Or are the reactions of the hydrates even faster than the non-hydrated glyoxal?

A7: Thanks for your suggestion and we regret for the incorrect expression. The hydrated form of carbonyls is more water-soluble than the un-hydrated form (EPA. 2012). Moreover, carbonyls are always in hydrate forms in aqueous reactions under atmospheric conditions (Liggio et al., 2005; Elrod et al., 2021; Michailoudi et al., 2021). But we think it's incorrect to directly compare the reactivity between dicarbonyls in monomer and in hydrate forms. We have deleted this sentence in our manuscript.

Q8: Page 7, line 199: "the most thermodynamically favored oligomer reactions for glyoxal and methylglyoxal" Specify that the reactions are for the hydrates, not (only) non-hydrated glyoxal and methylglyoxal.

A8: Thanks for your suggestion. We have specified that the reactions are for hydrates in our revised manuscript.

Q9: Page 8, line 208-209: "The product distribution of the reversible formation could well explain this phenomenon." How?

A9: We regret for the unclear expression of this sentences. We have rephrased the sentences in our revised manuscript.

C9: Lines 286-290 in Section 3.2

It increased significantly when RH increased from

Q11: Page 8, line 223: "Combined with the vapor pressure of dominant products" where do you get the vapor pressures of the dominant products?

A11: We get the vapor pressures of dominant products from previous studies. As for glyoxal, the vapor pressures of the reversible products are 10-6 atm and 10-11 atm for hydrates and oligomers, respectively (Hastings et al., 2005). And as for methylglyoxal, the vapor pressures of the reversible products are 10-5 atm and 10-11 atm for hydrates and oligomers, respectively (Axson et al., 2010). We have noted it in our revised manuscript.

C11: Lines 314-316 in Sect. 3.2:

Combined with the vapor pressure of dominant products published in previous studies (Hastings et al., 2005; Axson et al., 2010), their gas-particle partitioning coefficient can be roughly estimated and can effectively fit the field-measured values.

Q12: Figure 3: What are the lines in 3b? Also model like in 3c? Also, there are typos in the caption "(i) galyoxal and (ii) methylglyxoal".

A12: Thanks for your suggestion and we regret for the typos in the caption of Figure 2. The lines in Figure 3b are the fitting lines of irreversible uptake coefficients  $\gamma$  of dicarbonyls and SNA concentrations in aerosol liquid water. And we have corrected the typos in the caption in our revised manuscript.

Q13: Page 10, line 287: There are 2 figures in the Supplement labelled S7. In the second Fig. S7, what is the concentration unit for SNA in the ratios? Mass/mole/volume ratio?

A13: Thanks for your suggestion. We have revised the label of figures in the Supplement. And the concentration unit for SNA in the ratios is molality (mol/L ALWC). We have noted the concentration unit in our revised Supplement.

Q14: Page 13, line 369: "Furthermore, we note that there may be other potential explanations for the increase in particulate concentrations and the uncertainty in the gas-particle partitioning process." Particulate concentrations of what? And which partitioning processes?

A14: We regret for the unclear expression. "increase in Particulate concentrations" in this sentence refers to the increase in particle mass caused by dicarbonyls. And "partitioning processes" refers to all partitioning pathways, including physical adsorption, reversible pathways and irreversible pathways. We have specified this sentence in our revised manuscript.

C14: Lines 455-457 in Sect. 4:

Furthermore, we note that there may be other potential explanations for the increase in particle mass caused by dicarbonyls and the uncertainty in the gas-particle partitioning process, including physical adsorption, reversible pathways and irreversible pathways.

Q15: Page 13, line 371-372: "Other reversible pathways, like adducts formed from glyoxal with inorganic species, like sulfate and ammonia, could also promote the gasparticle partitioning process." I think you mean "such as", not "like". You used the word "like" similarly also earlier in the manuscript so check those too.

A15: Thanks for your suggestion. We have checked the word and revised it in our revised manuscript.

**Reference:**

- Axson, J. L., Takahashi, K., De Haan, D. O., and Vaida, V.: Gas-phase water-mediated equilibrium between methylglyoxal and its geminal diol, Proc Natl Acad Sci U S A, 107, 6687-6692, 10.1073/pnas.0912121107, 2010.
- Barsanti, K. C. and Pankow, J. F.: Thermodynamics of the formation of atmospheric organic particulate matter by accretion reactions—2. Dialdehydes, methylglyoxal, and diketones, Atmospheric Environment, 39, 6597-6607, 10.1016/j.atmosenv.2005.07.056, 2005.
- Chen, X., Zhang, Y., Zhao, J., Liu, Y., Shen, C., Wu, L., Wang, X., Fan, Q., Zhou, S., and Hang, J.: Regional modeling of secondary organic aerosol formation over eastern China: The impact of uptake coefficients of dicarbonyls and semivolatile process of primary organic aerosol, Science of the Total Environment, 793, 148176, 10.1016/j.scitotenv.2021.148176, 2021.
- Elrod, M. J., Sedlak, J. A., and Ren, H.: Accurate Computational Model for the Hydration Extent of Atmospherically Relevant Carbonyls on Aqueous Atmospheric Particles, ACS Earth and Space Chemistry, 5, 348-355, 10.1021/acsearthspacechem.0c00322, 2021.
- EPA., U.: Estimation Programs Interface Suite for Microsoft Windows, v 4.11., United States Environmental Protection Agency: Washington, DC, USA, https://www.epa.gov/tsca-screeningtools/epi-suitetm-estimation-programinterface., 2012.
- Hastings, W. P., Koehler, C. A., Bailey, E. L., and Haan, D. O. D.: Secondary organic aerosol formation by glyoxal hydration and oligomer formation: humidity effects and equilibrium shifts during analysis, Environmental Science & Technology, 39, 8728-8735, 2005.
- Healy, R. M., Temime, B., Kuprovskyte, K., and Wenger, J. C.: Effect of relative humidity on gas/particle partitioning and aerosol mass yield in the photooxidation of p-xylene, Environmental Science & Technology, 43, 1884-1889, 2009.
- Liggio, J., Shao-Meng, L. I., and Mclaren, R.: Heterogeneous Reactions of Glyoxal on Particulate Matter: Identification of Acetals and Sulfate Esters, Environmental Science & Technology, 39, 1532-1541, 2005.
- Michailoudi, G., Lin, J. J., Yuzawa, H., Nagasaka, M., Huttula, M., Kosugi, N., Kurtén, T., Patanen, M., and Prisle, N. L.: Aqueous-phase behavior of glyoxal and methylglyoxal observed with carbon and oxygen K-edge X-ray absorption spectroscopy, Atmospheric Chemistry and Physics, 21, 2881-2894, 10.5194/acp-21-2881-2021, 2021.
- Mitsuishi, K., Iwasaki, M., Takeuchi, M., Okochi, H., Kato, S., Ohira, S.-I., and Toda, K.: Diurnal Variations in Partitioning of Atmospheric Glyoxal and Methylglyoxal between Gas and Particles at the Ground Level and in the Free Troposphere, ACS Earth and Space Chemistry, 2, 915-924, 10.1021/acsearthspacechem.8b00037, 2018.
- Schweitzer, F., Magi, L., Mirabel, P., and George, C.: Uptake Rate Measurements of Methanesulfonic Acid and Glyoxal by Aqueous Droplets, Journal of Physical Chemistry A, 102, 593-600, 1998.

- Volkamer, R., San Martini, F., Molina, L. T., Salcedo, D., Jimenez, J. L., and Molina, M. J.: A missing sink for gas-phase glyoxal in Mexico City: Formation of secondary organic aerosol, Geophysical Research Letters, 34, 10.1029/2007gl030752, 2007.
- Xu, R., Li, X., Dong, H., Wu, Z., Chen, S., Fang, X., Gao, J., Guo, S., Hu, M., Li, D., Liu, Y., Liu, Y., Lou, S., Lu, K., Meng, X., Wang, H., Zeng, L., Zong, T., Hu, J., Chen, M., Shao, M., and Zhang, Y.: Measurement of gaseous and particulate formaldehyde in the Yangtze River Delta, China, Atmospheric Environment, 224, 10.1016/j.atmosenv.2019.117114, 2020.